# Fair Performance Metric Elicitation

**Gaurush Hiranandani**
UIUC
gaurush2@illinois.edu

**Harikrishna Narasimhan**
Google Research USA
hnarasimhan@google.com

**Oluwasanmi Koyejo**
UIUC & Google Research Accra
sanmi@illinois.edu

## Abstract

*What is a fair performance metric?* We consider the choice of fairness metrics through the lens of metric elicitation – a principled framework for selecting performance metrics that best reflect implicit preferences. The use of metric elicitation enables a practitioner to tune the performance and fairness metrics to the task, context, and population at hand. Specifically, we propose a novel strategy to elicit group-fair performance metrics for multiclass classification problems with multiple sensitive groups that also includes selecting the trade-off between predictive performance and fairness violation. The proposed elicitation strategy requires only relative preference feedback and is robust to both finite sample and feedback noise.

## 1  Introduction

Machine learning models are increasingly employed for critical decision-making tasks such as hiring and sentencing [44, 3, 11, 14, 31]. Yet, it is increasingly evident that automated decision-making is susceptible to bias, whereby decisions made by the algorithm are unfair to certain subgroups [5, 3, 10, 8, 31]. To this end, a wide variety of group fairness metrics have been proposed – all to reduce discrimination and bias from automated decision-making [25, 13, 17, 29, 49, 32]. However, a dearth of formal principles for selecting the most appropriate metric has highlighted the confusion of experts, practitioners, and end users in deciding which group fairness metric to employ [53]. This is further exacerbated by the observation that common metrics often lead to contradictory outcomes [29].

While the problem of selecting an appropriate fairness metric has gained prominence in recent years [17, 32, 53], it perhaps best understood as a special case of the task of choosing evaluation metrics in machine learning. For instance, when a cost-sensitive predictive model classifies patients into cancer categories [50] even without considering fairness, it is often unclear how the cost-tradeoffs be chosen so that they reflect the expert's decision-making, i.e., replacing expert intuition by quantifiable metrics. The recently proposed Metric Elicitation (ME) framework [20, 21] provides a solution. ME is a principled framework for eliciting performance metrics using feedback over classifiers from an end user. The motivation behind ME is that employing the performance metrics which reflect user tradeoffs will enable learning models that best capture user preferences [20]. As humans are often inaccurate in providing absolute preferences [41], Hiranandani et al. [20] propose to use pairwise comparison queries, where the user (oracle) is asked to compare two classifiers and provide a relative preference. Using such queries, ME aims to recover the oracle's metric. Figure 1 (reproduced from [20]) illustrates the ME framework.

Existing research suggests a fundamental trade-off between algorithmic fairness and performance [25, 51, 11, 7, 32, 53], where in addition to appropriate metrics, the practitioner or policymaker must choose a trade-off operating point between the competing objectives [53]. To this end, we extend the ME framework from eliciting multiclass classification metrics [21] to the task of eliciting *fair* performance metrics from pairwise preference feedback in the presence of multiple sensitive groups. In particular, we elicit metrics that reflect, jointly, the (i) predictive performance evaluated as a weighting of classifier's overall predictive rates, (ii) fairness violation assessed as the discrepancy in predictive rates among groups, and (iii) a trade-off between the predictive performance and fairness

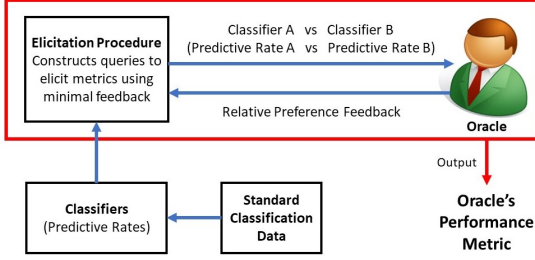

Figure 1: Framework of Metric Elicitation [20].

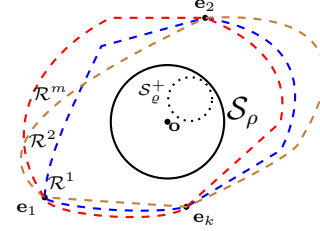

Figure 2: $\mathcal{R}^1 \times \cdots \times \mathcal{R}^m$ (best seen in colors); $\mathcal{R}^u \, \forall \, u \in [m]$ are convex sets with common vertices $\mathbf{e}_i \, \forall \, i \in [k]$ and enclose the sphere $\mathcal{S}_\rho$.

violation. Importantly, the elicited metrics are sufficiently flexible to encapsulate and generalize many existing predictive performance and fairness violation measures.

In eliciting group-fair performance metrics, we tackle three new challenges. First, from preference query perspective, the predictive performance and fairness violations are correlated, thus increasing the complexity of joint elicitation. Second, we find that in order to measure both positive and negative violations, the fair metrics are necessarily non-linear functions of the predictive rates, thus existing results on linear ME [21] cannot be applied directly. Finally, as we show, the number of groups directly impacts query complexity. We overcome these challenges by proposing a novel query efficient procedure that exploits the geometric properties of the set of rates.

**Contributions.** We consider metrics for algorithmically group-fair classification and propose a novel approach for eliciting predictive performance, fairness violations, and their trade-off point, from expert pairwise feedback. Our procedure uses binary-search based subroutines and recovers the metric with linear query complexity. Moreover, the procedure is robust to both finite sample and oracle feedback noise thus is useful in practice. Lastly, our method can be applied either by querying preferences over classifiers or rates. Such an equivalence is crucial for practical applications [20, 21].

**Notations.** Matrices and vectors are denoted by bold upper case and bold lower case letters, respectively. We denote the inner product of two vectors by $\langle \cdot, \cdot \rangle$ and the Hadamard product by $\odot$. The $\ell_2$-norm is denoted by $\|\cdot\|_2$. For $k \in \mathbb{Z}_+$, we represent the index set $\{1, 2, \cdots, k\}$ by $[k]$, and the $(k-1)$-dimensional simplex by $\Delta_k$. Given a matrix $\mathbf{A}$, $\textit{off-diag}(\mathbf{A})$ returns a vector of off-diagonal elements of $\mathbf{A}$ in row-major form. The group membership is denoted by superscripts and coordinates of vectors, matrices, and tuples are denoted by subscripts.

## 2 Background

The standard multiclass, multigroup classification setting comprises $k$ classes and $m$ groups with $X \in \mathcal{X}$, $G \in [m]$ and $Y \in [k]$ representing the input, group membership, and output random variables, respectively. The groups are assumed to be disjoint and known apriori [17, 29]. We have access to a dataset $\{(\mathbf{x}, g, y)_i\}_{i=1}^n$ of size $n$, generated *iid* from a distribution $\mathbb{P}(X, G, Y)$.

*Group-specific rates:* We consider separate (randomized) classifiers $h^g : \mathcal{X} \to \Delta_k$ for each group $g$, and use $\mathcal{H}^g = \{h^g : \mathcal{X} \to \Delta_k\}$ to denote the set of all classifiers for group $g$. The group-specific rate matrix $\mathbf{R}^g(h^g, \mathbb{P}) \in \mathbb{R}^{k \times k}$ for a classifier $h^g$ is given by:

$$R_{ij}^g(h^g, \mathbb{P}) := \mathbb{P}(h^g = j | Y = i, G = g) \quad \text{for } i, j \in [k]. \tag{1}$$

Since the diagonal entries of the rate matrix can be written in terms of the off-diagonal entries:

$$R_{ii}^g(h^g, \mathbb{P}) = 1 - \sum_{j=1, j \neq i}^{k} R_{ij}^g(h^g, \mathbb{P}), \tag{2}$$

any rate matrix is uniquely represented by its $q := (k^2 - k)$ off-diagonal elements as a vector $\mathbf{r}^g(h^g, \mathbb{P}) = \textit{off-diag}(\mathbf{R}^g(h^g, \mathbb{P}))$. So we will interchangeably refer to the rate matrix as a *'vector of rates'*. The feasible set of rates associated with a group $g$ is denoted by $\mathcal{R}^g = \{\mathbf{r}^g(h^g, \mathbb{P}) : h^g \in \mathcal{H}^g\}$. For clarity, we will suppress the dependence on $\mathbb{P}$ and $h^g$ if it is clear from the context.

*Overall rates:* We define the overall classifier $h : (\mathcal{X}, [m]) \to \Delta_k$ by $h(\mathbf{x}, g) := h^g(\mathbf{x})$ and denote its tuple of group-specific rates by:

$$\mathbf{r}^{1:m} := (\mathbf{r}^1, \ldots, \mathbf{r}^m) \in \mathcal{R}^1 \times \cdots \times \mathcal{R}^m =: \mathcal{R}^{1:m}.$$

This tuple allows us to measure the fairness violation across groups. The fairness violation is believed to be in trade-off with the predictive performance [25, 7, 32]. The latter is measured using the overall rate matrix of the classifier $h$:

$$R_{ij} := \mathbb{P}(h = j | Y = i) = \sum_{g=1}^{m} t_i^g R_{ij}^g, \qquad (3)$$

where $t_i^g := \mathbb{P}(G = g | Y = i)$ is the prevalence of group $g$ within class $i$. For an overall classifier $h$, the *'vector of rates'* $\mathbf{r} = \textit{off-diag}(\mathbf{R})$ can be conveniently written in terms of its group-specific tuple of rates as $\mathbf{r} = \sum_{g=1}^{m} \boldsymbol{\tau}^g \odot \mathbf{r}^g$, where $\boldsymbol{\tau}^g := \textit{off-diag}([\mathbf{t}^g \ \mathbf{t}^g \dots \mathbf{t}^g])$.

*Fairness violation measure:* The (approximate) fairness of a classifier is often determined by the 'discrepancy' in rates across different groups e.g. *equalized odds* [17, 4]. So given two groups $u, v \in [m]$, we define the discrepancy in their rates as:

$$\mathbf{d}^{uv} := |\mathbf{r}^u - \mathbf{r}^v|. \qquad (4)$$

Since there are $m$ groups, the number of *discrepancy vectors* are $\binom{m}{2}$ .

## 2.1 Fair Performance Metric

We aim to elicit a general class of metrics, which recovers and generalizes existing fairness measures, based on trade-off between predictive performance and fairness violation [25, 17, 10, 7, 32]. Let $\phi : [0,1]^q \to \mathbb{R}$ be the cost of overall misclassification (aka. predictive performance) and $\varphi : [0,1]^{m \times q} \to \mathbb{R}$ be the fairness violation cost for a classifier $h$ determined by the overall rates $\mathbf{r}(h)$ and group discrepancies $\{\mathbf{d}^{uv}(h)\}_{u,v=1,v>u}^m$, respectively. Without loss of generality (wlog), we assume the metrics $\phi$ and $\varphi$ are costs. Moreover, the metrics are scale invariant as global scale does not affect the learning problem [36]; hence let $\phi : [0,1]^q \to [0,1]$ and $\varphi : [0,1]^{m \times q} \to [0,1]$.

**Definition 1.** *Fair Performance Metric: Let $\phi$ and $\varphi$ be monotonically increasing linear functions of overall rates and group discrepancies, respectively. The fair metric $\Psi$ is a trade-off between $\phi$ and $\varphi$. In particular, given $\mathbf{a} \in \mathbb{R}^q, \mathbf{a} \geq 0$ (misclassification weights), a set of vectors $\mathbf{B} := \{\mathbf{b}^{uv} \in \mathbb{R}^q, \mathbf{b}^{uv} \geq 0\}_{u,v=1,v>u}^m$ (fairness violation weights), and a scalar $\lambda$ (trade-off) with*

$$\|\mathbf{a}\|_2 = 1, \qquad \sum_{u,v=1,v>u}^{m} \|\mathbf{b}^{uv}\|_2 = 1, \qquad 0 \leq \lambda \leq 1, \qquad (5)$$

*(wlog., due to scale invariance), we define the metric $\Psi$ as:*

$$\Psi(\mathbf{r}^{1:m}; \mathbf{a}, \mathbf{B}, \lambda) := \underbrace{(1-\lambda)}_{\textit{trade-off}} \underbrace{\langle \mathbf{a}, \mathbf{r} \rangle}_{\phi(\mathbf{r})} + \lambda \underbrace{\left( \sum_{u,v=1,v>u}^{m} \langle \mathbf{b}^{uv}, \mathbf{d}^{uv} \rangle \right)}_{\varphi(\mathbf{r}^{1:m})}. \qquad (6)$$

Examples of the misclassification cost $\phi(\mathbf{r})$ include cost-sensitive linear metrics [1]. Many existing fairness metrics for two classes and two groups such as *equal opportunity* [17], *balance for the negative class* [29] *error-rate balance* (i.e., $0.5|r_1^1 - r_1^2| + 0.5|r_2^1 - r_2^2|$) [10], *weighted equalized odds* (i.e., $b_1|r_1^1 - r_1^2| + b_2|r_2^1 - r_2^2|$) [17, 7], etc. correspond to fairness violations of the form $\varphi(\mathbf{r}^{1:m})$ considered above. The combination of $\phi(\mathbf{r})$ and $\varphi(\mathbf{r}^{1:m})$ as defined in $\Psi(\mathbf{r}^{1:m})$ appears regularly in prior work [25, 7, 32]. Notice that the metric is flexible to allow different fairness violation costs for different pairs of groups thus capable of enabling reverse discrimination [38]. Lastly, while the metric is linear with respect to (wrt.) the discrepancies, it is non-linear wrt. the group-wise rates. Hence, standard linear ME algorithm [21] cannot be trivially applied for eliciting the metric in Definition 1.

## 2.2 Fair Performance Metric Elicitation; Problem Statement

We now state the problem of *Fair Performance Metric Elicitation (FPME)* and define the associated *oracle query*. The broad definitions follow from Hiranandani et al. [20, 21], extended so the rates and the performance metrics correspond to the multiclass multigroup-fair classification setting.

**Definition 2** (Oracle Query). *Given two classifiers $h_1, h_2$ (equivalent to a tuple of rates $\mathbf{r}_1^{1:m}, \mathbf{r}_2^{1:m}$ respectively), a query to the Oracle (with metric $\Psi$) is represented by:*

$$\Gamma(h_1, h_2) = \Omega\left(\mathbf{r}_1^{1:m}, \mathbf{r}_2^{1:m}\right) = \mathbb{1}[\Psi(\mathbf{r}_1^{1:m}) > \Psi(\mathbf{r}_2^{1:m})], \qquad (7)$$

*where $\Gamma : \mathcal{H} \times \mathcal{H} \to \{0,1\}$ and $\Omega : \mathcal{R}^{1:m} \times \mathcal{R}^{1:m} \to \{0,1\}$. In words, the query asks whether $h_1$ is preferred to $h_2$ (equivalent to whether $\mathbf{r}_1^{1:m}$ is preferred to $\mathbf{r}_2^{1:m}$), as measured by $\Psi$.*

In practice, the oracle can be an expert, a group of experts, or an entire user population. The ME framework can be applied by posing classifier comparisons directly to them via interpretable learning techniques [42, 12] or via A/B testing [45]. For example, one may perform A/B testing for an internet-based application by deploying two classifiers A and B and use the population's level of engagement to decide the preference between the two classifiers. For other applications, intuitive visualizations of the predictive rates for two different classifiers (see e.g., [53, 6]) can be used to ask preference feedback from a group of domain experts.

We emphasize that the metric $\Psi$ used by the oracle is unknown to us and can be accessed only through queries to the oracle. Since the metrics we consider are functions of rates, comparing two classifiers on a metric is equivalent to comparing their corresponding rates. Henceforth, we will denote any query to the oracle by a pair of rates $(\mathbf{r}_1^{1:m}, \mathbf{r}_2^{1:m})$. Also, whenever we refer to an oracles's dimension, we are referring to the dimension of its rate arguments. For instance, we will consider the oracle in Definition 2 to be of dimension $m \times q$. Next, we formally state the FPME problem.

**Definition 3** (Fair Performance Metric Elicitation with Pairwise Queries (given $\{(\mathbf{x}, g, y)_i\}_{i=1}^n$)). *Suppose that the oracle's (unknown) performance metric is $\Psi$. Using oracle queries of the form $\Omega(\hat{\mathbf{r}}_1^{1:m}, \hat{\mathbf{r}}_2^{1:m})$, where $\hat{\mathbf{r}}_1^{1:m}, \hat{\mathbf{r}}_2^{1:m}$ are the estimated rates from samples, recover a metric $\hat{\Psi}$ such that $\|\Psi - \hat{\Psi}\| < \omega$ under a suitable norm $\|\cdot\|$ for sufficiently small error tolerance $\omega > 0$.*

Similar to the standard metric elicitation problems [20, 21], the performance of FPME is evaluated both by the fidelity of the recovered metric and the query complexity. As done in decision theory literature [30, 20], we present our FPME solution by first assuming access to population quantities such as the population rates $\mathbf{r}^{1:m}(h, \mathbb{P})$, and then discuss how elicitation can be performed from finite samples, e.g., with empirical rates $\hat{\mathbf{r}}^{1:m}(h, \{(\mathbf{x}, g, y)_i\}_{i=1}^n)$.

### 2.3 Linear Performance Metric Elicitation – Warmup

We revisit the Linear Performance Metric Elicitation (LPME) procedure from [21], which we will use as as a subroutine to elicit fair performance metrics. The LPME procedure assumes an enclosed sphere $\mathcal{S} \subset \mathcal{Z}$, where $\mathcal{Z}$ is the $q$-dimensional space of classifier statistics that are feasible, i.e., can be achieved by some classifier. It also assumes access to a $q$-dimensional oracle $\Omega'$ whose scale invariant linear metric is of the form $\xi(\mathbf{z}) := \langle \mathbf{a}, \mathbf{z} \rangle$ with $\|\mathbf{a}\|_2 = 1$, analogous to the misclassification cost in Definition 1. Analogously, the oracle queries are of the type $\Omega'(\mathbf{z}_1, \mathbf{z}_2) := \mathbb{1}[\xi(\mathbf{z}_1) > \xi(\mathbf{z}_2)]$.

When the number of classes $k = 2$, LPME elicits the coefficients $\mathbf{a}$ using a simple one-dimensional binary search. When $k > 2$, LPME performs binary search in each coordinate while keeping the others fixed, and performs this in a coordinate-wise fashion until convergence. By restricting this coordinate-wise binary search procedure to posing queries from within a sphere $\mathcal{S}$, LPME can be equivalently seen as minimizing a strongly-convex function and shown to converge to a solution $\hat{\mathbf{a}}$ close to $\mathbf{a}$. Specifically, the algorithm takes the query space $\mathcal{S} \subset \mathcal{Z}$, binary-search tolerance $\epsilon$, and the oracle $\Omega'$ as input, and by querying $O(q \log(1/\epsilon))$ queries recovers $\hat{\mathbf{a}}$ with $\|\hat{\mathbf{a}}\|_2 = 1$ such that $\|\mathbf{a} - \hat{\mathbf{a}}\|_2 \leq O(\sqrt{q}\epsilon)$ (Theorem 2 in [21]). We provide details of the LPME procedure in Algorithm 2 (Appendix A) for completeness and summarize the discussion with the following remark.

**Remark 1.** *Given a $q$-dimensional space $\mathcal{Z}$ enclosing a sphere $\mathcal{S} \subset \mathcal{Z}$ and an oracle $\Omega'$ with linear metric $\xi(\mathbf{z}) := \langle \mathbf{a}, \mathbf{z} \rangle$, the LPME algorithm (Algorithm 2, Appendix A) provides an estimate $\hat{\mathbf{a}}$ with $\|\hat{\mathbf{a}}\|_2 = 1$ such that the estimated slope is close to the true slope, i.e., $a_i/a_j \approx \hat{a}_i/\hat{a}_j \ \forall \ i, j \in [q]$.*

Note that the algorithm estimates the direction of the coefficient vector, not its magnitude.

## 3 Geometry of the product set $\mathcal{R}^{1:m}$

The LPME procedure described above works with rate queries of dimension $q$. We would like to use this procedure to elicit the fair metrics in Definition 1 defined on tuples of dimension $m \times q$. So to make use of LPME, we restrict our queries to a $q$-dimensional sphere $\mathcal{S}$ which is common to the feasible rate region $\mathcal{R}^g$ for each group $g$, i.e. to a sphere in the intersection $\mathcal{R}^1 \cap \ldots \cap \mathcal{R}^m$. We show now that such a sphere does indeed exist under a mild assumption.

**Assumption 1.** *For all groups, the conditional-class distributions are not identical, i.e., $\forall \ g \in [m], \forall \ i \neq j, \mathbb{P}(Y = i | X, G = g) \neq \mathbb{P}(Y = j | X, G = g)$. In other words, there is some non-trivial signal for classification for each group.*

Let $\mathbf{e}_i \in \{0, 1\}^q$ be the rate profile for a trivial classifier that predicts class $i$ on all inputs. Note that these trivial classifiers evaluate to the same rates $\mathbf{e}_i$ irrespective of which group we apply them to.

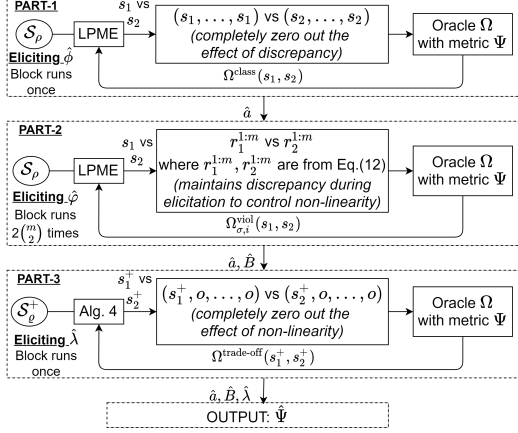

| **Algorithm 1: FPM Elicitation** |
|---|
| **Input:** Query spaces $\mathcal{S}_\rho$, $\mathcal{S}_\varrho^+$, search tolerance $\epsilon > 0$, and oracle $\Omega$ |
| 1:    $\hat{\mathbf{a}} \leftarrow \text{LPME}(\mathcal{S}_\rho, \epsilon, \Omega^{\text{class}})$ |
| 2:    **If** $m == 2$ |
| 3:      $\check{\mathbf{f}} \leftarrow \text{LPME}(\mathcal{S}_\rho, \epsilon, \Omega_1^{\text{viol}})$ |
| 4:      $\tilde{\mathbf{f}} \leftarrow \text{LPME}(\mathcal{S}_\rho, \epsilon, \Omega_2^{\text{viol}})$ |
| 5:      $\hat{\mathbf{b}}^{12} \leftarrow$ normalized solution from (11) |
| 6:    **Else** Let $\mathcal{L} \leftarrow \varnothing$ |
| 7:      **For** $\sigma \in \mathcal{M}$ **do** |
| 8:        $\check{\mathbf{f}}^\sigma \leftarrow \text{LPME}(\mathcal{S}_\rho, \epsilon, \Omega_{\sigma,1}^{\text{viol}})$ |
| 9:        $\tilde{\mathbf{f}}^\sigma \leftarrow \text{LPME}(\mathcal{S}_\rho, \epsilon, \Omega_{\sigma,k}^{\text{viol}})$ |
| 10:       Let $\ell^\sigma$ be Eq. (13), extend $\mathcal{L} \leftarrow \mathcal{L} \cup \{\ell^\sigma\}$ |
| 11:      $\hat{\mathbf{B}} \leftarrow$ normalized solution from (14) using $\mathcal{L}$ |
| 12:    $\hat{\lambda} \leftarrow$ Algorithm 4 $(\mathcal{S}_\varrho^+, \epsilon, \Omega^{\text{trade-off}})$ |
| **Output:** $\hat{\mathbf{a}}, \hat{\mathbf{B}}, \hat{\lambda}$ |

Figure 3: Workflow of the FPME procedure.

**Proposition 1** (Geometry of $\mathcal{R}^{1:m}$; Figure 2). *For any group $g \in [m]$, the set of confusion rates $\mathcal{R}^g$ is convex, bounded in $[0,1]^q$, and has vertices $\{\mathbf{e}_i\}_{i=1}^k$. The intersection of group rate sets $\mathcal{R}^1 \cap \cdots \cap \mathcal{R}^m$ is convex and always contains the rate $\mathbf{o} = \frac{1}{k}\sum_{i=1}^k \mathbf{e}_i$ in the interior, which is associated with the uniform random classifier that predicts each class with equal probability.*

Since $\mathcal{R}^1 \cap \cdots \cap \mathcal{R}^m$ is convex and always contains a point $\mathbf{o}$ in the interior, we can make the following remark (see Figure 2 for an illustration).

**Remark 2** (Existence of common sphere $\mathcal{S}_\rho$). *There exists a $q$-dimensional sphere $\mathcal{S}_\rho \subset \mathcal{R}^1 \cap \cdots \cap \mathcal{R}^m$ of non-zero radius $\rho$ centered at $\mathbf{o}$. Thus, any rate $\mathbf{s} \in \mathcal{S}_\rho$ is feasible for all groups, i.e., $\mathbf{s}$ is achievable by some classifier $h^g$ for all groups $g \in [m]$.*

A method to obtain $\mathcal{S}_\rho$ with suitable radius $\rho$ from [21] is discussed in Appendix B.1. From Remark 2, we observe that any tuple of group rates $\mathbf{r}^{1:m} = (\mathbf{s}^1, \dots, \mathbf{s}^m)$ chosen from $\mathcal{S}_\rho \times \dots \times \mathcal{S}_\rho$ is achievable for some choice of group-specific classifiers $h^1, \dots, h^m$. Moreover, when two groups $u, v$ are assigned the same rate profile $\mathbf{s} \in \mathcal{S}_\rho$, the fairness discrepancy $\mathbf{d}^{uv} = \mathbf{0}$. We will exploit these observations in the elicitation strategy we discuss next.

## 4    Metric Elicitation

We have access to an oracle whose (unknown) metric $\Psi$ given in Definition 1 is parameterized by $(\mathbf{a}, \mathbf{B}, \lambda)$. The proposed FPME framework for eliciting the oracle's metric is presented in Figure 3 and is summarized in Algorithm 1. The procedure has three parts executed in sequence: (a) eliciting the misclassification cost $\phi(\mathbf{r})$ (i.e., $\mathbf{a}$), (b) eliciting the fairness violation $\varphi(\mathbf{r}^{1:m})$ (i.e., $\mathbf{B}$), and (c) eliciting the trade-off between the misclassification cost and fairness violation (i.e., $\lambda$). For simplicity, we will suppress the coefficients $(\mathbf{a}, \mathbf{B}, \lambda)$ from the notation $\Psi$ whenever it is clear from context.

Notice that the metric $\Psi$ is *piece-wise linear* in its coefficients. So our high level idea is to restrict the queries we pose to the oracle to lie within regions where the metric $\Psi$ is linear, so that we can then employ the LPME subroutine to elicit the corresponding linear coefficients. We will show for each of the three components (a)–(c), how we can identify regions in the query space where the metric is linear and apply the LPME procedure (or a variant of it). By restricting the query inputs to those regions, we will essentially be converting the $(m \times q)$-dimensional oracle $\Omega$ in Definition 2 into an equivalent $q$-dimensional oracle that compares rates $\mathbf{s}_1, \mathbf{s}_2$ from the common sphere $\mathcal{S}_\rho \subset \mathcal{R}^1 \cap \cdots \cap \mathcal{R}^m$. We first discuss our approach assuming the oracle has no *feedback* noise, and later in Section 5 show that our approach is robust to noisy feedback and provide query complexity guarantees.

### 4.1    Eliciting the Misclassification Cost $\phi(\mathbf{r})$: Part 1 in Figure 3 and Line 1 in Algorithm 1

To elicit the misclassification cost coefficients $\mathbf{a}$, we will query from a region of the query space where the fairness violation term in the metric is zero. Specifically, we will query group rate profile of the form $\mathbf{r}^{1:m} = (\mathbf{s}, \dots, \mathbf{s})$, where $\mathbf{s}$ is a $q$-dimensional rate from the common sphere $\mathcal{S}_\rho$. For these group rate profiles, the metric $\Psi$ simply evaluates to the linear misclassification term, i.e.:

$$\Psi(\mathbf{s}, \dots, \mathbf{s}) = (1 - \lambda)\langle \mathbf{a}, \mathbf{s} \rangle.$$

So given a pair of group rate profiles $\mathbf{r}_1^{1:m} = (\mathbf{s}_1, \dots, \mathbf{s}_1)$ and $\mathbf{r}_2^{1:m} = (\mathbf{s}_2, \dots, \mathbf{s}_2)$, where $\mathbf{s}_1, \mathbf{s}_2 \in \mathcal{S}_\rho$, the oracle's response will essentially compare $\mathbf{s}_1$ and $\mathbf{s}_2$ on the linear metric $(1 - \lambda)\langle \mathbf{a}, \mathbf{s} \rangle$. Hence, we estimate the coefficients $\mathbf{a}$ by applying LPME over the $q$-dimensional sphere $\mathcal{S}_\rho$ with a modified oracle $\Omega^{\text{class}}$ which takes a pair of rate profiles $\mathbf{s}_1$ and $\mathbf{s}_2$ from $\mathcal{S}_\rho$ as input, and responds with:

$$\Omega^{\text{class}}(\mathbf{s}_1, \mathbf{s}_2) = \Omega((\mathbf{s}_1, \dots, \mathbf{s}_1), (\mathbf{s}_2, \dots, \mathbf{s}_2)).$$

This is decribed in line 1 of Algorithm 1, which applies the LPME subroutine with query space $\mathcal{S}_\rho$, binary search tolerance $\epsilon$, and the oracle $\Omega^{\text{class}}$. From Remark 1, this subroutine returns a coefficient vector $\mathbf{f}$ with $\|\mathbf{f}\|_2 = 1$ such that:

$$\frac{(1 - \lambda)a_i}{(1 - \lambda)a_j} = \frac{f_i}{f_j} \implies \frac{a_i}{a_j} = \frac{f_i}{f_j}. \tag{8}$$

By setting $\hat{\mathbf{a}} = \mathbf{f}$, we recover the classification coefficients independent of the fairness violation coefficients and trade-off parameter. See part 1 in Figure 3 for further illustration.

## 4.2 Eliciting the Fairness Violation $\varphi(\mathbf{r}^{1:m})$: Part 2 in Figure 3 and lines 2-11 in Algorithm 1

We now discuss eliciting the fairness term $\varphi(\mathbf{r}^{1:m})$. We will first discuss the special case of $m = 2$ groups and later discuss how the proposed procedure can be extended to handle multiple groups.

### 4.2.1 Special Case of $m = 2$: Lines 2-5 in Algorithm 1

Recall from Definition 1 that in the violation term, we measure the group discrepancies using the *absolute* difference between the group rates, i.e. $\mathbf{d}^{12} = |\mathbf{r}^1 - \mathbf{r}^2|$. If we restrict our queries to only those rate profiles $\mathbf{r}^{1:2}$ for which the difference in each coordinate of $\mathbf{r}^1 - \mathbf{r}^2$ is either always positive or always negative, then we can treat the violation term as a linear metric within this region and apply LPME to estimate the associated coefficients.

To this end, we pose to the oracle queries of the form $\mathbf{r}^{1:2} = (\mathbf{s}, \mathbf{e}_i)$, where we assign to group 1 a rate profile $\mathbf{s}$ from the common sphere $\mathcal{S}_\rho$, and to group 2 the rate profile $\mathbf{e}_i \in \{0, 1\}^q$ for some $i$. Remember that $\mathbf{e}_i$ is a rate vector associated with a trivial classifier which predicts class $i$ on all inputs, and is therefore a binary vector. Since we know whether an entry of $\mathbf{e}_i$ is either a 0 or a 1, we can decipher the signs of each entry of the difference vector $\mathbf{s} - \mathbf{e}_i$. Hence for group rate profiles of the above form, the metric $\Psi$ can be written as a linear function in $\mathbf{s}$:

$$\Psi(\mathbf{s}, \mathbf{e}_i) = \langle (1 - \lambda)\mathbf{a} \odot (\mathbf{1} - \boldsymbol{\tau}^2) + \lambda \mathbf{w}_i \odot \mathbf{b}^{12}, \mathbf{s} \rangle + c_i, \tag{9}$$

where $\mathbf{w}_i := \mathbf{1} - 2\mathbf{e}_i$ tells us the sign of each entry of $\mathbf{s} - \mathbf{e}_i$, $c_i$ is a constant, and we have used the fact that $\boldsymbol{\tau}^1 = \mathbf{1} - \boldsymbol{\tau}^2$. Fixing a class $i$, we then apply LPME over the $q$-dimensional sphere $\mathcal{S}_\rho$ with a modified oracle $\Omega_i^{\text{viol}}$ which takes a pair of rate profiles $\mathbf{s}_1, \mathbf{s}_2 \in \mathcal{S}_\rho$ as input and responds with:

$$\Omega_i^{\text{viol}}(\mathbf{s}_1, \mathbf{s}_2) = \Omega((\mathbf{s}_1, \mathbf{e}_i), (\mathbf{s}_2, \mathbf{e}_i)). \tag{10}$$

One run of LPME with oracle $\Omega_1^{\text{viol}}$ results in $q - 1$ independent equations. In order to elicit a $q$-dimensional vector $\mathbf{b}^{12}$, we must run LPME again with oracle $\Omega_2^{\text{viol}}$. This is described in lines 3 and 4 of Algorithm 1. The LPME calls provide us with two slopes $\check{\mathbf{f}}, \tilde{\mathbf{f}}$ such that $\|\check{\mathbf{f}}\|_2 = \|\tilde{\mathbf{f}}\|_2 = 1$ from which it is easy to obtain the fairness violation weights:

$$\hat{\mathbf{b}}^{12} = \frac{\tilde{\mathbf{b}}^{12}}{\|\tilde{\mathbf{b}}^{12}\|_2}, \quad \text{with} \quad \tilde{\mathbf{b}}^{12} = \mathbf{w}_1 \odot \left[ \delta \check{\mathbf{f}} - \hat{\mathbf{a}} \odot (\mathbf{1} - \boldsymbol{\tau}^2) \right], \tag{11}$$

where $\delta$ is a scalar depending on the known entities $\boldsymbol{\tau}^{12}, \hat{\mathbf{a}}, \check{\mathbf{f}}^{12}, \tilde{\mathbf{f}}^{12}$. The derivation is provided in Appendix C.2.1. Because $\varphi$ is scale invariant (see Definition 1), the normalized solution $\hat{\mathbf{b}}^{12}$ is independent of the true trade-off $\lambda$ and depends only on the previously elicited vector $\hat{\mathbf{a}}$.

### 4.2.2 General Case of $m > 2$: Lines 6-11 in Algorithm 1

We briefly outline the elicitation procedure for $m > 2$ groups, with details in Appendix C.2.2. Let $\mathcal{M}$ be a set of subsets of the $m$ groups such that each element $\sigma \in \mathcal{M}$ and $[m] \setminus \sigma$ partition the set of $m$ groups. We will later discuss how to choose $\mathcal{M}$ for efficient elicitation. Similar to the two-group case, we pose queries $\mathbf{r}^{1:m}$ where to a subset of groups $\sigma \in \mathcal{M}$, we assign the trivial rate vector $\mathbf{e}_i$ and to the rest $[m] \setminus \sigma$ groups, we assign a point $\mathbf{s}$ from the common sphere $\mathcal{S}_\rho$. Observe that within

this query region, the metric $\Psi$ is linear in its inputs. So for a fixed partitioning of groups defined by $\sigma$, we apply LPME with a query space $\mathcal{S}_\rho$ using the modified $q$-dimensional oracle:

$$\Omega^{\text{viol}}_{\sigma,i}(\mathbf{s}_1, \mathbf{s}_2) = \Omega(\mathbf{r}_1^{1:m}, \mathbf{r}_2^{1:m}) \text{ where } \mathbf{r}_1^g = \begin{cases} \mathbf{e}_i & \text{if } g \in \sigma \\ \mathbf{s}_1 & \text{o.w.} \end{cases} \text{ and } \mathbf{r}_2^g = \begin{cases} \mathbf{e}_i & \text{if } g \in \sigma \\ \mathbf{s}_2 & \text{o.w.} \end{cases}. \quad (12)$$

As described in lines 8 and 9 of the algorithm, we repeat this twice fixing class $i$ to 1 and $k$. The guarantees for LPME then give us the following relationship between coefficients $\mathbf{b}^{uv}$ we wish to elicit and the already elicited coefficient $\hat{\mathbf{a}}$:

$$\sum_{u,v} \mathbb{1}\left[|\{u,v\} \cap \sigma| = 1\right] \tilde{\mathbf{b}}^{uv} = \mathbf{w}_1 \odot \left[\delta^\sigma \breve{\mathbf{f}}^\sigma - \hat{\mathbf{a}} \odot (\mathbf{1} - \boldsymbol{\tau}^\sigma)\right], \quad (13)$$

where $\boldsymbol{\tau}^\sigma = \sum_{g \in \sigma} \boldsymbol{\tau}^g$ and $\tilde{\mathbf{b}}^{uv} := \lambda \mathbf{b}^{uv}/(1-\lambda)$ is a scaled version of the true (unknown) $\mathbf{b}^{uv}$. Since we need to estimate $\binom{m}{2}$ coefficients, we repeat the above procedure for $\binom{m}{2}$ partitions of the groups defined by $\sigma$ and get a system of $\binom{m}{2}$ linear equations. We may choose any $\mathcal{M}$ of size $\binom{m}{2}$ so that the equations are independent. From the solution to these equations, we recover $\tilde{\mathbf{b}}^{uv}$'s, which we normalize to get estimates of the final fairness violation weights:

$$\hat{\mathbf{b}}^{uv} = \frac{\tilde{\mathbf{b}}^{uv}}{\sum_{u,v=1,v>u}^{m} \|\tilde{\mathbf{b}}^{uv}\|_2} \quad \text{for} \quad u, v \in [m], v > u. \quad (14)$$

Thanks to the normalization, the elicited fairness violation weights are independent of the trade-off $\lambda$.

### 4.3 Eliciting Trade-off $\lambda$: Part 3 in Figure 3 and Line 12 in Algorithm 1

Equipped with estimates of the misclassification and fairness violation coefficients $(\hat{\mathbf{a}}, \hat{\mathbf{B}})$, the final step is to elicit the trade-off $\lambda$ between them. We now show how this can be posed as one-dimensional binary search problem. Suppose we restrict our queries to be of the form $\mathbf{r}^{1:m} = (\mathbf{s}^+, \mathbf{o}, \ldots, \mathbf{o})$, where for all but the first group, we assign the rate $\mathbf{o}$ associated with a uniform random classifier, and for the first group, we assign some rate $\mathbf{s}^+$ such that $\mathbf{s}^+ \geq \mathbf{o}$. For these rate profiles, the group rate difference terms $\mathbf{r}^1 - \mathbf{r}^v = \mathbf{s}^+ - \mathbf{o} \geq \mathbf{0}$ for all $v \in \{2, \ldots, m\}$, and all the other difference terms are $\mathbf{0}$. As a result, the metric $\Psi$ is linear in the input rate profiles:

$$\Psi(\mathbf{s}^+, \mathbf{o}, \ldots, \mathbf{o}) = \langle (1-\lambda)\boldsymbol{\tau}^1 \odot \mathbf{a} + \lambda \sum_{v=2}^{m} \mathbf{b}^{1v}, \mathbf{s}^+ \rangle + c, \quad (15)$$

where $c$ is a constant. Despite the metric being linear in the identified input region, we cannot directly apply the LPME procedure described in Section 2.3 to elicit $\lambda$, because we have one parameter to elicit but the input to the metric is $q$-dimensional. Here we propose a slight variant of LPME.

Similar to the original procedure [20], we first construct a one-dimensional function $\vartheta$, which takes a guess of the trade-off parameter as input, and outputs the quality of the guess. We show that this function is unimodal and its mode coincides with the oracle's true trade-off parameter $\lambda$.

**Lemma 1.** *Let $\mathcal{S}_\varrho^+ \subset \mathcal{S}_\rho$ be a $q$-dimensional sphere with radius $\varrho < \rho$ such that $\mathbf{s}^+ \geq \mathbf{o}, \forall \mathbf{s}^+ \in \mathcal{S}_\varrho^+$ (see Figure 2). Assume the estimates $\hat{\mathbf{a}}$ and $\hat{\mathbf{b}}^{uv}$'s satisfy a mild regularity condition $\langle \hat{\mathbf{a}}, \sum_{v=2}^{m} \hat{\mathbf{b}}^{1v} \rangle \neq 1$. Define a one-dimensional function $\vartheta$ as:*

$$\vartheta(\bar{\lambda}) := \Psi(\mathbf{s}_{\bar{\lambda}}^*, \mathbf{o}, \ldots, \mathbf{o}), \quad (16)$$

*where*

$$\mathbf{s}_{\bar{\lambda}}^* = \underset{\mathbf{s}^+ \in \mathcal{S}_\varrho^+}{\operatorname{argmax}} \langle (1-\bar{\lambda})\boldsymbol{\tau}^1 \odot \hat{\mathbf{a}} + \bar{\lambda} \sum_{v=2}^{m} \hat{\mathbf{b}}^{1v}, \mathbf{s}^+ \rangle. \quad (17)$$

*Then the function $\vartheta$ is strictly quasiconcave (and therefore unimodal) in $\bar{\lambda}$. Moreover, the mode of this function is achieved at the oracle's true trade-off parameter $\lambda$.*

For a candidate trade-off $\bar{\lambda}$, the function $\vartheta$ first constructs a candidate linear metric based on (15), maximizes this candidate metric over inputs $\mathbf{s}^+$, and evaluates the oracle's true metric $\Psi$ at the maximizing rate profile. Note that we cannot directly compute the function $\vartheta$ as it needs the oracle's metric $\Psi$. However, given two candidates for the trade-off parameter $\bar{\lambda}_1$ and $\bar{\lambda}_2$, one can compare the values of $\vartheta(\bar{\lambda}_1)$ and $\vartheta(\bar{\lambda}_2)$ by finding the corresponding maximizers over $\mathbf{s}^+$ and querying the oracle to compare them. Because $\vartheta$ is unimodal, one can use a simple binary search using such pairwise comparisons to find the mode of the function, which we know coincides with the true $\lambda$. We provide an outline of this procedure in Algorithm 4 in Appendix C.3, which uses the modified oracle

$$\Omega^{\text{trade-off}}(\mathbf{s}_1^+, \mathbf{s}_2^+) = \Omega((\mathbf{s}_1^+, \mathbf{o}, \ldots, \mathbf{o}), (\mathbf{s}_2^+, \mathbf{o}, \ldots, \mathbf{o}))$$

to compare the maximizers in (17). We also discuss in Appendix C.3 how the maximizer in (17) can be computed efficiently. Combining parts 1, 2 and 3 in Figure 3 completes the FPME procedure.

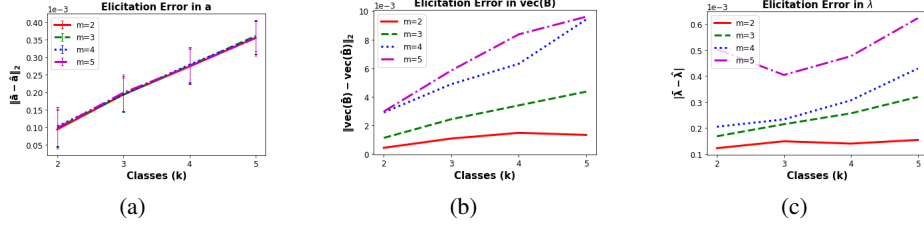

Figure 4: Elicitation error in recovering the oracle's metric.

## 5 Guarantees

We discuss elicitation guarantees under the following feedback model.

**Definition 4** (Oracle Feedback Noise: $\epsilon_\Omega \geq 0$). *For two rates $\mathbf{r}_1^{1:m}, \mathbf{r}_2^{1:m} \in \mathcal{R}^{1:m}$, the oracle responds correctly as long as $|\Psi(\mathbf{r}_1^{1:m}) - \Psi(\mathbf{r}_2^{1:m})| > \epsilon_\Omega$. Otherwise, it may be incorrect.*

In words, the oracle may respond incorrectly if the rates are very close as measured by the metric $\Psi$. Since deriving the final metric involves offline computations including certain ratios, we discuss guarantees under a regularity assumption that ensures all components are well defined.

**Assumption 2.** *We assume that $1 > c_1 > \lambda > c_2 > 0$, $\min_i |a_i| > c_3$, $\min_i |(1 - \lambda)a_i\tau_i^\sigma - \lambda w_{ji}b_i^\sigma| > c_4 \, \forall \, j \in [q], \sigma \in \mathcal{M}$, for some $c_1, c_2, c_3, c_4 > 0$, $\rho > \varrho \gg \epsilon_\Omega$, and $\langle \mathbf{a}, \sum_{v=2}^m \mathbf{b}^{1v} \rangle \neq 1$.*

**Theorem 1.** *Given $\epsilon, \epsilon_\Omega \geq 0$, and a 1-Lipschitz fair performance metric $\Psi$ parametrized by $\mathbf{a}, \mathbf{B}, \lambda$, under Assumptions 1 and 2, Algorithm 1 returns a metric $\hat{\Psi}$ with parameters:*

- $\hat{\mathbf{a}}$ : *after $O\left(q \log \frac{1}{\epsilon}\right)$ queries such that $\|\mathbf{a} - \hat{\mathbf{a}}\|_2 \leq O\left(\sqrt{q}(\epsilon + \sqrt{\epsilon_\Omega/\rho})\right)$.*

- $\hat{\mathbf{B}}$ : *after $O\left(\binom{m}{2} q \log \frac{1}{\epsilon}\right)$ queries such that $\|vec(\mathbf{B}) - vec(\hat{\mathbf{B}})\|_2 \leq O\left(mq(\epsilon + \sqrt{\epsilon_\Omega/\rho})\right)$, where $vec(\cdot)$ vectorizes the matrix.*

- $\hat{\lambda}$ : *after $O(\log(\frac{1}{\epsilon}))$ queries, with error $|\lambda - \hat{\lambda}| \leq O\left(\epsilon + \sqrt{\epsilon_\Omega/\varrho} + \sqrt{mq(\epsilon + \sqrt{\epsilon_\Omega/\rho})/\varrho}\right)$.*

We see that the proposed FPME procedure is robust to noise, and its query complexity depends linearly in the number of unknown entities. For instance, line 1 in Algorithm 1 elicits $\hat{\mathbf{a}} \in \mathbf{R}^q$ by posing $\tilde{O}(q)$ queries, the 'for' loop in line 7 of Algorithm 1 runs for $\binom{m}{2}$ iterations, where each iteration requires $\tilde{O}(2q)$ queries, and finally line 12 in Algorithm 1 is a simple binary search requiring $\tilde{O}(1)$ queries. Previous work suggests that linear multiclass elicitation (LPME) elicits misclassification costs ($\phi$) with linear query complexity [21]. Surprisingly, our proposed FPME procedure elicits a more complex (nonlinear) metric without increasing the query complexity order. Furthermore, since sample estimates of rates are consistent estimators, and the metrics discussed are 1-Lipschitz wrt. rates, with high probability, we gather correct oracle feedback from querying with finite sample estimates $\Omega(\hat{\mathbf{r}}_1^{1:m}, \hat{\mathbf{r}}_2^{1:m})$ instead of querying with population statistics $\Omega(\mathbf{r}_1^{1:m}, \mathbf{r}_2^{1:m})$, as long as we have sufficient samples. Apart from this, Algorithm 1 is agnostic to finite sample errors as long as the sphere $\mathcal{S}_\rho$ is contained within the feasible region $\mathcal{R}^1 \cap \cdots \cap \mathcal{R}^m$.

## 6 Experiments

We first empirically validate the FPME procedure and recovery guarantees of Section 5. Recall that there exists a sphere $\mathcal{S}_\rho \subset \mathcal{R}^1 \cap \cdots \cap \mathcal{R}^m$ as long as there is a non-trivial classification signal within each group (Assumption 2). Thus for experiments, we assume access to a feasible sphere $\mathcal{S}_\rho$ with $\rho = 0.2$. We randomly generate 100 oracle metrics each for $k, m \in \{2, 3, 4, 5\}$ parametrized by $\{\mathbf{a}, \mathbf{B}, \lambda\}$. This specifies the query outputs by the oracle for each metric in Algorithm 1. We then use Algorithm 1 with tolerance $\epsilon = 10^{-3}$ to elicit corresponding metrics parametrized by $\{\hat{\mathbf{a}}, \hat{\mathbf{B}}, \hat{\lambda}\}$. Algorithm 1 makes $1 + 2M$ subroutine calls to LPME procedure and 1 call to Algorithm 4. LPME subroutine requires exactly $16(q-1)\log(\pi/2\epsilon)$ queries, where we use 4 queries to shrink the interval in the binary search loop and fix 4 cycles for the coordinate-wise search. Also, Algorithm 4 requires $4\log(1/\epsilon)$ queries. In Figure 4, we report the mean of the $\ell_2$-norm between the oracle's metric and the elicited metric. Clearly, we elicit metrics that are close to the true metrics. Moreover, this holds true across a range of $m$ and $k$ values demonstrating the robustness of the proposed approach. Figure 4(a)

shows that the error $\|\mathbf{a} - \hat{\mathbf{a}}\|_2$ increases only with the number of classes $k$ and not groups $m$. This is expected since $\hat{\mathbf{a}}$ is elicited by querying rates that zero out the fairness violation (Section 4.1). Figure 4(b) verifies Theorem 1 by showing that $\|\text{vec}(\mathbf{B}) - \text{vec}(\hat{\mathbf{B}})\|_2$ increases with both number of classes $k$ and groups $m$. In accord with Theorem 1, Figure 4(c) shows that the elicited trade-off $\hat{\lambda}$ is also close to the true $\lambda$. However, the elicitation error increases consistently with groups $m$ but not with classes $k$. A possible reason may be the cancellation of errors from eliciting $\hat{\mathbf{a}}$ and $\hat{\mathbf{B}}$ separately.

We also highlight the utility of FPME in ranking real-world classifiers in Appendix E. We find that even though FPME may make elicitation error as discussed above, it achieves near perfect ranking of classifiers; whereas, the default baseline metrics fail to achieve good rankings (see Appendix E for details). To connect to practice, this implies that when given a set of classifiers, ranking based on elicited metrics will align most closely to ranking based on the true metric, as compared to ranking classifiers based on default metrics. This is a crucial advantage of ME for practical purposes.

## 7 Related Work

Some early attempts to eliciting individual fairness metrics [22, 33] are distinct from ours – as we are focused on the more prevalent setting of group fairness, yet for which there are no existing approaches to our knowledge. Zhang et al. [53] propose an approach that elicits only the trade-off between accuracy and fairness using complicated ratio queries. We, on the other hand, elicit classification cost, fairness violation, and the trade-off together as a non-linear function, all using much simpler pairwise comparison queries. Prior work for constrained classification focus on learning classifiers under constraints for fairness [16, 17, 52, 34]. We take the regularization view of algorithmic fairness, where a fairness violation is embedded in the metric definition instead of as constraints [25, 7, 11, 2, 32]. From the elicitation perspective, the closest line of work to ours is Hiranandani et al. [20, 21], who propose the problem of ME but solve it only for a simpler setting of classification without fairness. As we move to multiclass, multigroup fair performance ME, we find that the complexity of both the form of the metrics and the query space increases. This results in starkly different elicitation strategy with novel methods required to provide query complexity guarantees. Learning (linear) functions passively using pairwise comparisons is a mature field [24, 19, 40], but these approaches fail to control sample (i.e. query) complexity. Active learning in fairness [37] is a related direction; however the aim there is to learn a fair classifier based on fixed metric instead of eliciting the metric itself.

## 8 Discussion Points and Future Work

- **Transportability:** Our elicitation procedure is independent of the population $\mathbb{P}$ as long as there exists a sphere of rates which is feasible for all groups. Thus, any metric that is learned using one dataset or model class (i.e. by estimated $\hat{\mathbb{P}}$) can be applied to other applications and datasets, as long as the expert believes the context and tradeoffs are the same.

- **Extensions.** Our propsal can be modified to leverage the structure in the metric or the groups to further reduce the query complexity. For example, when the fairness violation weights are the same for all pairs of groups, the procedure in Section 4.2.2 requires only one partitioning of groups to elicit the metric $\hat{\varphi}$. Such modifications are easy to incorporate. In the future, we plan to extend our approach to more complex metrics such as linear-fractional functions of rates and discrepancies.

- **Limitations of group-fair metrics.** Since the metrics we consider depend on a classifier only through its rates, comparing two classifiers on these metrics is equivalent to comparing their rates. Unfortunately, with this setup, all the limitations associated with group-fairness definition of metrics apply to our setup as well. For example, we may discard notions of *individual fairness* when only group-rates are considered for comparing classifiers [9]. Similarly, issues associated with *overlapping groups* [27], *detailed group specification* [27], *unknown or changing groups* [18, 15], *noisy or biased* group information [48], among others, pose limitations to our proposed setup. We hope that as the first work on the topic, our work will inspire the research community to address many of these open problems for the task of metric elicitation.

- **Optimal bounds.** We conjecture that our query complexity bounds are tight; however, we leave this detail for the future. In conclusion, we elicit a more complex (non-linear) group fair-metric with the same query complexity order as standard classification linear elicitation procedures [21].

## Broader Impact

Machine Learning community has constructed many methods that create bias-free classifiers; however, it has been long accepted that fairness is not only an algorithmic concept but a societal one [53, 10]. So machine learning systems cannot presume how policymakers would like to handle fairness but have to elicit fairness criteria from them. Thus our primary contribution is a framework for selecting metrics that can be tuned to expert panel preferences and are designed to measure intrinsic fairness tradeoffs. We hope that by eliciting metrics, algorithmic fairness can be better tuned to the tradeoffs of stakeholders, or be used to compare preferences of different possible stakeholders. This paper seeks to truly democratize and personalize fair machine learning. Besides, the significance of fair performance metric elicitation lies in how it empowers the practitioner to tune the design of machine learning models to the needs of the target fairness task. The 'transportability' aspect of the proposed procedure is crucial here since it allows practitioners to elicit metrics using any estimated distribution, perhaps from simple model class, and then use the elicited fair metric for optimizing complex models or evaluating models in test time for the target fairness task.

At the same time, this work may have drawbacks because it leaves open the key question of who should be the stakeholders to be queried. This work also assumes a parametric form for the oracle metric, which may not be an exact match to practice. Furthermore, we should be cautious of the result of the failure of the system which could cause disparate impact among sensitive groups when the elicited metric is incorrect, e.g., when applied to settings where the stated assumptions are not met.

## Acknowledgments and Disclosure of Funding

We thank the anonymous reviewers for providing helpful and constructive feedback on the paper. Funding in support of this work: None.

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
