[Supplementary Material]

# Appendices

## A  Linear Performance Metric Elicitation

As explained in Section 2.3, we use the linear metric elicitation procedure [21] as a subroutine in order to elicit a more complicated metric as defined in Definition 1. For completeness, we provide the details here.

The linear metric elicitation procedure proposed in [21] assumes an enclosed sphere $\mathcal{S} \subset \mathcal{Z}$, where $\mathcal{Z}$ is the $q$-dimensional space of classifier statistics that are feasible, i.e., can be achieved by some classifier. Let the the radius of the sphere $\mathcal{S}$ be $\rho$. We extend the linear metric elicitation procedure (Algorithm 2 in [21]) to elicit any linear metric (without the monotonicity condition) defined over the space $\mathcal{Z}$. This is because in Section 4.2, we require to elicit slopes that are not necessarily for monotonic metrics (e.g., see Equation (9)). Let the oracle's scale invariant metric be $\xi(\mathbf{z}) := \langle \mathbf{a}, \mathbf{z} \rangle$, such that $\|\mathbf{a}\|_2 = 1$. Analogously, the oracle queries are $\Omega'(\mathbf{z}_1, \mathbf{z}_2) := \mathbb{1}[\xi(\mathbf{z}_1) > \xi(\mathbf{z}_2)]$. We start by outlining a trivial Lemma from [21].

**Lemma 2.** *[21] Let $\xi$ be a linear metric parametrized by $\mathbf{a}$ such that $\|\mathbf{a}\|_2 = 1$, then the unique optimal classifier statistic $\mathbf{z}$ over the sphere $\mathcal{S}$ is a point on the boundary of $\mathcal{S}$ given by $\mathbf{z} = \rho\mathbf{a} + \mathbf{o}$, where $\mathbf{o}$ is the center of the sphere $\mathcal{S}$.*

Given a linear performance metric, Lemma 2 provides a unique point in the query space which lies on the boundary of the sphere $\partial\mathcal{S}$. Moreover, the converse also holds true that given a point on the boundary of the sphere $\partial\mathcal{S}$, one may recover the linear metric for which the given point is optimal. Thus, in order to elicit a linear metric, Hiranandani et al. [21] essentially search for the optimal statistic (over the surface of the sphere) using pairwise queries to the oracle which in turn reveals the true metric. The algorithm is summarized in Algorithm 2. The algorithm also uses the following standard paramterization for the surface of the sphere $\partial\mathcal{S}$.

**Parameterizing the boundary of the enclosed sphere** $\partial\mathcal{S}$**.** Let $\boldsymbol{\theta}$ be a $(q-1)$-dimensional vector of angles, where all the angles except the primary angle are in $[0, \pi]$, and the primary angle is in $[0, 2\pi]$. A linear performance metric with $\|\mathbf{a}\|_2 = 1$ is constructed by setting $a_i = \Pi_{j=1}^{i-1} \sin\theta_j \cos\theta_i$ for $i \in [q-1]$ and $a_q = \Pi_{j=1}^{q-1} \sin\theta_j$. By using Lemma 2, the metric's optimal classifier statistic over the sphere $\mathcal{S}$ is easy to compute. Thus, varying $\boldsymbol{\theta}$ in this procedure, parametrizes the surface of the sphere $\partial\mathcal{S}$. We denote this parametrization by $\mu(\boldsymbol{\theta})$, where $\mu : [0, \pi]^{q-2} \times [0, 2\pi] \to \partial\mathcal{S}$.

*Description of Algorithm 2:*[1] Suppose that the oracle's linear metric is $\xi$ parametrized by $\mathbf{a}$ where $\|\mathbf{a}\|_2 = 1$ (Section 2.3). Using the parametrization $\mu(\boldsymbol{\theta})$ of the surface of the sphere $\partial\mathcal{S}$ as explained above, Algorithm 2 returns an estimate $\hat{\mathbf{a}}$ with $\|\hat{\mathbf{a}}\|_2 = 1$. Line 2-6 in Algorithm 2 recovers the orthant of the optimal statistic over the sphere by posing $q$ trivial queries. Once the search orthant of the optimal statistic is fixed, the procedure is same as Algorithm 2 of [21]. In each iteration of the for loop, the algorithm updates one angle $\theta_j$ keeping other angles fixed by a binary-search procedure, where the *ShrinkInterval* subroutine (illustrated in Figure 5) shrinks the interval $[\theta_j^a, \theta_j^b]$ by half based on the responses. Then the algorithm cyclically updates each angle until it converges to a metric sufficiently close to the true metric. The number of cycles in coordinate-wise search is fixed to four.

## B  Proofs and Details of Section 3

*Proof of Proposition 1.* The set of rates $\mathcal{R}^g$ for a group $g$ satisfies the following properties:

- *Convex*: Let us take two classifiers $h_1^g, h_2^g \in \mathcal{H}^g$ which achieve the rates $\mathbf{r}_1^g, \mathbf{r}_2^g \in \mathcal{R}^g$. We need to check whether or not the convex combination $\alpha\mathbf{r}_1^g + (1-\alpha)\mathbf{r}_2^g$ is feasible, i.e., there exists some classifier which achieve this rate. Consider a classifier $h^g$, which with probability $\alpha$ predicts what classifier $h_1^g$ predicts and with probability $1-\alpha$ predicts what classifier $h_2^g$ predicts. Then the elements of the rate matrix $R_{ij}^g(h)$ is given by:

**Algorithm 2** Linear Performance Metric Elicitation
---

1: **Input:** Query space $\mathcal{S}$, binary-search tolerance $\epsilon > 0$, oracle $\Omega'$ with metric $\xi$

2: **for** $i = 1, 2, \cdots q$ **do**
3:     Set $\mathbf{a} = \mathbf{a}' = (1/\sqrt{q}, \ldots, 1/\sqrt{q})$.
4:     Set $a'_i = -1/\sqrt{q}$.
5:     Compute the optimal $\bar{z}^{(\mathbf{a})}$ and $\bar{z}^{(\mathbf{a}')}$ over the sphere $\mathcal{S}$ using Lemma 2
6:     Query $\Omega'(\mathbf{z}^{(\mathbf{a})}, \mathbf{z}^{(\mathbf{a}')})$
    {Fix the search orthant based on the above oracle responses}

7: **Initialize:** $\boldsymbol{\theta} = \boldsymbol{\theta}^{(1)}$                     {$\boldsymbol{\theta}^{(1)}$ is any point in the search orthant.}
8: **for** $t = 1, 2, \cdots, T = 4(q-1)$ **do**
9:     Set $\boldsymbol{\theta}^{(a)} = \boldsymbol{\theta}^{(c)} = \boldsymbol{\theta}^{(d)} = \boldsymbol{\theta}^{(e)} = \boldsymbol{\theta}^{(b)} = \boldsymbol{\theta}^{(t)}$.
10:     **while** $\left|\theta_j^{(b)} - \theta_j^{(a)}\right| > \epsilon$ **do**
11:        Set $\theta_j^{(c)} = \frac{3\theta_j^{(a)} + \theta_j^{(b)}}{4}, \theta_j^{(d)} = \frac{\theta_j^{(a)} + \theta_j^{(b)}}{2}$, and $\theta_j^{(e)} = \frac{\theta_j^{(a)} + 3\theta_j^{(b)}}{4}$.
12:        Set $\mathbf{z}^{(a)} = \mu(\boldsymbol{\theta}^{(a)})$ (i.e. parametrization of $\partial\mathcal{S}$). Similarly, set $\mathbf{z}^{(c)}, \mathbf{z}^{(d)}, \mathbf{z}^{(e)}, \mathbf{z}^{(b)}$
13:        Query $\Omega'(\mathbf{z}^{(c)}, \mathbf{z}^{(a)}), \Omega'(\mathbf{z}^{(d)}, \mathbf{z}^{(c)}), \Omega'(\mathbf{z}^{(e)}, \mathbf{z}^{(d)}), \Omega'(\mathbf{z}^{(b)}, \mathbf{z}^{(e)})$.
14:        $[\theta_j^{(a)}, \theta_j^{(b)}] \leftarrow$ *ShrinkInterval* (responses)           {see Figure 5}
15:     Set $\theta_j^{(d)} = \frac{1}{2}(\theta_j^{(a)} + \theta_j^{(b)})$
16:     Set $\boldsymbol{\theta}^{(t)} = \boldsymbol{\theta}^{(d)}$.
17: **Output:** $\hat{a}_i = \Pi_{j=1}^{i-1} \sin\theta_j^{(T)} \cos\theta_i^{(T)} \, \forall i \in [q-1], \; \hat{a}_q = \Pi_{j=1}^{q-1} \sin\theta_j^{(T)}$

---

**Subroutine *ShrinkInterval***

**Input:** Oracle responses for $\Omega'(\mathbf{z}^{(c)}, \mathbf{z}^{(a)})$, $\Omega'(\mathbf{z}^{(d)}, \mathbf{z}^{(c)}), \Omega'(\mathbf{z}^{(e)}, \mathbf{z}^{(d)}), \Omega'(\mathbf{z}^{(b)}, \mathbf{z}^{(e)})$

**If** $(\mathbf{z}^{(a)} \succ \mathbf{z}^{(c)})$ Set $\theta_j^{(b)} = \theta_j^{(d)}$.

**elseif** $(\mathbf{z}^{(a)} \prec \mathbf{z}^{(c)} \succ \mathbf{z}^{(d)})$ Set $\theta_j^{(b)} = \theta_j^{(d)}$.

**elseif** $(\mathbf{z}^{(c)} \prec \mathbf{z}^{(d)} \succ \mathbf{z}^{(e)})$ Set $\theta_j^{(a)} = \theta_j^{(c)}, \theta_j^{(b)} = \theta_j^{(e)}$.

**elseif** $(\mathbf{z}^{(d)} \prec \mathbf{z}^{(e)} \succ \mathbf{z}^{(b)})$ Set $\theta_j^{(a)} = \theta_j^{(d)}$.

**else** Set $\theta_j^{(a)} = \theta_j^{(d)}$.

**Output:** $[\theta_j^{(a)}, \theta_j^{(b)}]$.

Figure 5: (Left): Subroutine *ShrinkInterval*. (Right): Visual intuition of the subroutine *ShrinkInterval* [21]; the subroutine shrinks the current interval to half based on oracle responses to the four queries.

$$
\begin{aligned}
R_{ij}^g(h) &= \mathbb{P}(h^g = j | Y = i) \\
&= \mathbb{P}(h_1^g = j | h^g = h_1^g, Y = i)\mathbb{P}(h^g = h_1^g) + \mathbb{P}(h_2^g = j | h^g = h_2^g, Y = i)\mathbb{P}(h^g = h_2^g) \\
&= \alpha \mathbf{r}_1^g + (1 - \alpha)\mathbf{r}_2^g.
\end{aligned}
$$

Therefore, $\mathcal{R}^g \; \forall \; g \in [m]$ is convex.

- *Bounded:* Since $R_{ij}^g(h) = P[h = j | Y = i] = P[h = j, Y = i]/P[Y = i] \leq 1$ for all $i, j \in [k]$, $\mathcal{R}^g \subseteq [0, 1]^q$.

- $\mathbf{e}_i$'s and $\mathbf{o}$ are always achieved: The classifier which always predicts class $i$, will achieve the rate $\mathbf{e}_i$. Thus, $\mathbf{e}_i \in \mathcal{R}^g \, \forall i \in [k], g \in [m]$ are feasible. Just like the convexity proof, a classifier which predicts similar to one of the trivial classifiers with probability $1/k$ will achieve the rates $\mathbf{o}$.

- $\mathbf{e}_i$'s are vertices: Any supporting hyperplane with slope $\ell_{1i} < \ell_{1j} < 0$ and $\ell_{1p} = 0$ for $p \in [k], p \neq i, j$ will be supported by $\mathbf{e}_1$ (corresponding to the trivial classifier which predict class 1). Thus, $\mathbf{e}_i$'s are vertices of the convex set. As long as the class-conditional distributions are not identical, i.e., there is some signal for non-trivial classification conditioned on each group [21], one can construct a ball around the trivial rate $\mathbf{o}$ and thus $\mathbf{o}$ lies in the interior. $\qquad\qquad\qquad\qquad\qquad\qquad\qquad\qquad\qquad\qquad\qquad\qquad\qquad\qquad\qquad\qquad\square$

---

**Algorithm 3** Obtaining the sphere $\mathcal{S}_\rho$ with radius $\rho$

---

1: **Input:** The center $\mathbf{o}$ of the feasible region of rates across groups.
2: **for** $j = 1, 2, \cdots, q$ **do**
3:    Let $\mathbf{r}_j$ be the standard basis vector for the $j$-th dimension.
4:    Compute the maximum $\ell_j$ such that $\mathbf{o} + \ell_j \mathbf{r}_j$ is feasible for all groups by solving (OP1).
5: Let $CONV$ be the convex hull of $\{\mathbf{o} \pm \ell_j \mathbf{r}_j\}_{j=1}^q$.
6: Compute the radius $s$ of the largest ball which can fit inside of $CONV$, centered at $\mathbf{o}$.
7: **Output:** Sphere $\mathcal{S}_\rho$ with radius $\rho = s$ centered at $\mathbf{o}$.

---

## B.1 Finding the Sphere $\mathcal{S}_\rho$

In this section, we discuss how a sufficiently large sphere $\mathcal{S}_\rho$ with radius $\rho$ may be found. The following discussion is extended from [21] to multiple groups setting and provided here for completeness.

The following optimization problem is a special case of OP2 in [34, 46]. The problem corresponds to feasiblity check problem for a given rate $\mathbf{r}_0$ achieved by all groups within small error $\epsilon > 0$.

$$\min_{\mathbf{r}^g \in \mathcal{R}^g \ \forall g \in [m]} 0 \qquad s.t. \ \|\mathbf{r}^g - \mathbf{r}_0\|_2 \leq \epsilon \quad \forall \, g \in [m]. \tag{OP1}$$

The above problem checks the feasibility and if a solution to the above problem exists, then Algorithm 1 of [34] returns it. The approach in [34] constructs a classifier whose group-wise rates are $\epsilon$-close to the given rate $\mathbf{r}_0$.

Furthermore, Algorithm 3 computes a value of $\rho \geq \tilde{s}/k$, where $\tilde{s}$ is the radius of the largest ball contained in the set $\mathcal{R}^1 \cap \cdots \cap \mathcal{R}^m$. Notice that the approach in [34] is consistent, thus we should get a good estimate of the sphere, provided we have sufficient samples. The algorithm runs offline and does not impact query complexity.

**Lemma 3.** *[21] Let $\tilde{s}$ be the radius of the largest ball centered at $\mathbf{o}$ in $\mathcal{R}^1 \cap \cdots \cap \mathcal{R}^m$. Then Algorithm 3 returns a radius $\rho \geq \tilde{s}/k$.*

*Proof.* Let $\ell_j$ be as computed in the algorithm and $\ell := \min_j \ell_j$, then we have $\ell \geq \tilde{s}$. Moreover, the region $CONV$ contains the convex hull of $\{o \pm \ell \mathbf{e}_j\}_{j=1}^q$; however, this region contains a ball of radius $\ell/\sqrt{q} = \ell/\sqrt{k^2 - k} \geq \ell/k \geq \tilde{s}/k$, and thus $\rho \geq \tilde{s}/k$. □

## C Derivations of Section 4

Notice that $\sum_{g=1}^m \boldsymbol{\tau}^g = \mathbf{1}$, i.e., the vector of ones.

### C.1 Eliciting the Misclassification Cost $\phi(\mathbf{r})$; Part 1 in Figure 3 and line 1 in Algorithm 1

The key to eliciting $\phi$ is to remove the effect of fairness violation $\varphi$ in the oracle responses. As explained in Section 4.1, we run the LPME procedure (Algorithm 2) with the $q$-dimensional query space $\mathcal{S}_\rho$, binary search tolerance $\epsilon$, the equivalent oracle $\Omega^{\text{class}}$. From Remark 1, this subroutine returns a slope $\mathbf{f}$ with $\|\mathbf{f}\|_2 = 1$ such that:

$$\frac{(1 - \lambda)a_i}{(1 - \lambda)a_j} = \frac{f_i}{f_j} \implies \frac{a_i}{a_j} = \frac{f_i}{f_j}. \tag{18}$$

Thus, we set $\hat{\mathbf{a}} := \mathbf{f}$ (line 1, Algorithm 1).

### C.2 Eliciting the Fairness Violation $\varphi(\mathbf{r}^{1:m})$; Part 2 in Figure 3 and lines 2-11 in Algorithm 1

#### C.2.1 Eliciting the Fairness Violation $\varphi(\mathbf{r}^{1:m})$ for $m = 2$; lines 2-5 in Algorithm 1

For $m = 2$, we have only one vector of unfairness weights $\mathbf{b}^{12}$, which we now aim to elicit given $\hat{\mathbf{a}}$. As discussed in Section 4.2.1, we fix trivial rates (through trivial classifiers) to one group and allow non-trivial rates from $\mathcal{S}_\rho$ on another group. This essentially makes the metric in Definition 1 linear. The elicitation procedure is as follows.

Fix trivial classifier predicting class 1 for group 2 i.e. fix $h^2(x) = 1 \,\forall\, x \in \mathcal{X}$, and thus $\mathbf{r}^2 = \mathbf{e}_1$. For group 1, we constrain the confusion rates to lie in the sphere $\mathcal{S}_\rho$ i.e. $\mathbf{r}^1 = \mathbf{s}$ for $\mathbf{s} \in \mathcal{S}_\rho$. Then the metric in Definition 1 amounts to:

$$\Psi((\mathbf{s}, \mathbf{e}_1); \mathbf{a}, \mathbf{b}^{12}, \lambda) = (1 - \lambda)\langle \mathbf{a} \odot (1 - \boldsymbol{\tau}^2), \mathbf{s}\rangle + \lambda\langle \mathbf{b}^{12}, |\mathbf{e}_1 - \mathbf{s}|\rangle + c_1. \tag{19}$$

The above is a function of $\mathbf{s} \in \mathcal{S}_\rho$. Since $\mathbf{e}_i$'s are binary vectors and since $0 \le \mathbf{s} \le 1$, the sign of the absolute function with respect to $\mathbf{s}$ can be recovered. Recall that the rates are defined in row major form of the rate matrices, thus $\mathbf{e}_1$ is 1 at every $(k + j * (k - 1))$-th coordinate, where $j \in \{0, \dots, k - 2\}$, and 0 otherwise. The coordinates where the confusion rates are 1 in $\mathbf{e}_1$, the absolute function opens with a negative sign (wrt. $\mathbf{s}$) and with a positive sign otherwise. In particular, define a $q$-dimensional vector $\mathbf{w}_1$ with entries $-1$ at every $(k + j * (k - 1))$-th coordinate, where $j \in \{0, \dots, k - 2\}$, and 1 otherwise. One may then write the metric $\Psi$ as:

$$\Psi((\mathbf{s}, \mathbf{e}_1)\,;\, \mathbf{a}, \mathbf{b}^{12}, \lambda) = \langle (1 - \lambda)\mathbf{a} \odot (\mathbf{1} - \boldsymbol{\tau}^2) + \lambda\mathbf{w}_1 \odot \mathbf{b}^{12}, \mathbf{s}\rangle + c_1. \tag{20}$$

This is again a linear metric elicitation problem where $\mathbf{s} \in \mathcal{S}$. We may again use the LPME procedure (Algorithm 2), which outputs a (normalized) slope $\check{\mathbf{f}}$ with $\|\check{\mathbf{f}}\|_2 = 1$ in line 3 of Algorithm 1. Using Remark 1, we get $q - 1$ independent equations and may represent every element of $\mathbf{b}^{12}$ based on one element, say $\bar{b}_{k-1}^{12}$, i.e.:

$$\frac{\check{f}_{k-1}}{\check{f}_i} = \frac{(1 - \lambda)(1 - \tau_{k-1}^2)\bar{a}_{k-1} + \lambda\bar{b}_{k-1}^{12}}{(1 - \lambda)(1 - \tau_i^2)\bar{a}_i + \lambda w_{1i}\bar{b}_i^{12}} \qquad \forall\, i \in [q].$$

$$\implies \lambda\mathbf{b}^{12} = \mathbf{w}_1 \odot \left[ \left( \frac{(1 - \lambda)(1 - \tau_{k-1}^2)\bar{a}_{k-1} + \lambda\bar{b}_{k-1}^{12}}{\check{f}_{k-1}} \right) \check{\mathbf{f}} - (1 - \lambda)((1 - \boldsymbol{\tau}^2) \odot \mathbf{a}) \right]. \tag{21}$$

In order to elicit entire $\mathbf{b}^{12}$, we need one more linear relation such as (21). So, we now fix the trivial classifier predicting class $k$ for group 2 i.e. fix $h^2(x) = k \,\forall\, \mathbf{x} \in \mathcal{X}$, and thus $\mathbf{r}^2 = \mathbf{e}_k$. For group 1, we constrain the rates to again lie in the sphere $\mathcal{S}_\rho$ i.e. $\mathbf{r}^1 = \mathbf{s}$ for $\mathbf{s} \in \mathcal{S}_\rho$. Since the rate vectors are in row major form of the rate matrices, notice that $\mathbf{e}_k$ is 1 at every $(k - 1 + j * (k - 1))$-th coordinate, where $j \in \{0, \dots, k - 2\}$, and 0 otherwise. In particular, define a $q$-dimensional vector $\mathbf{w}_k$ with entries $-1$ at every $(k - 1 + j * (k - 1))$-th coordinate, where $j \in \{0, \dots, k - 2\}$, and 1 otherwise. One may then write the metric $\Psi$ as:

$$\Psi((\mathbf{s}, \mathbf{e}_k); \mathbf{a}, \mathbf{b}^{12}, \lambda) = (1 - \lambda)\langle \mathbf{a} \odot (1 - \boldsymbol{\tau}^2), \mathbf{s}\rangle + \lambda\langle \mathbf{b}^{12}, |\mathbf{e}_k - \mathbf{s}|\rangle + c_k. \tag{22}$$

This is a linear metric elicitation problem where $\mathbf{s} \in \mathcal{S}$. Thus, line 4 of Algorithm 1 applies LPME subroutine (Algorithm 2), which outputs a slope $\tilde{\mathbf{f}}$ with $\|\tilde{\mathbf{f}}\|_2 = 1$. Using Remark 1, we extract the following relation between two of its coordinates, say the $(k - 1)$-th and $((k - 1)^2 + 1)$-th coordinates:

$$\frac{\tilde{f}_{k-1}}{\tilde{f}_{(k-1)^2+1}} = \frac{(1 - \lambda)(1 - \tau_{k-1}^2)\bar{a}_{k-1} - \lambda\bar{b}_{k-1}^{12}}{(1 - \lambda)(1 - \tau_{(k-1)^2+1}^2)\bar{a}_{(k-1)^2+1} + \lambda\bar{b}_{(k-1)^2+1}^{12}}. \tag{23}$$

Combining equations (21) and (23) and replacing the true $\mathbf{a}$ with the estimated $\hat{\mathbf{a}}$ from Section 4.1, we have an estimate of the scaled substitute as:

$$\tilde{\mathbf{b}}^{12} = \mathbf{w}_1 \odot \left[ \delta\check{\mathbf{f}}^{12} - \hat{\mathbf{a}} \odot (\mathbf{1} - \boldsymbol{\tau}^2) \right], \tag{24}$$

$$\text{where } \delta = \frac{2(1 - \tau_{k-1}^2)\hat{a}_{k-1}}{\check{f}_{k-1}} \left[ \frac{\frac{(1 - \tau_{(k-1)^2+1}^2)\hat{a}_{(k-1)^2+1}}{(1 - \tau_{k-1}^2)\hat{a}_{k-1}} - \frac{\tilde{f}_{(k-1)^2+1}}{\tilde{f}_{k-1}}}{\left( \frac{\check{f}_{(k-1)^2+1}}{\check{f}_{k-1}} - \frac{\tilde{f}_{(k-1)^2+1}}{\tilde{f}_{k-1}} \right)} \right]$$

and $\tilde{\mathbf{b}}$ is a scaled substitute defined as $\tilde{\mathbf{b}}^{12} := \frac{\lambda}{(1-\lambda)}\mathbf{b}^{12}$, which nonetheless is computable from (24). Since we require a solution $\hat{\mathbf{b}}$ such that $\|\hat{\mathbf{b}}\|_2 = 1$ (Definition 1), we normalize $\tilde{\mathbf{b}}$ and get the final solution:

$$\hat{\mathbf{b}}^{12} = \frac{\tilde{\mathbf{b}}^{12}}{\|\tilde{\mathbf{b}}^{12}\|_2}. \tag{25}$$

Notice that, due to the above normalization, the solution is independent of the true trade-off $\lambda$.

## C.2.2 Eliciting the Fairness Violation $\varphi(\mathbf{r}^{1:m})$ for $m > 2$; line 6-11 in Algorithm 1

Consider a non-empty set of sets $\mathcal{M} \subset 2^{[m]} \setminus \{\varnothing, [m]\}$. We will later discuss how to choose $\mathcal{M}$ for efficient elicitation. When $m > 2$, we partition the set of groups $[m]$ into two sets of groups. Let $\sigma \in \mathcal{M}$ and $[m] \setminus \sigma$ be one such partition of the $m$ groups defined by the set $\sigma$. We follow exactly similar procedure as in the previous section i.e. fixing trivial rates (through trivial classifiers) on the groups in $\sigma$ and allowing non-trivial rates from $\mathcal{S}_\rho$ on the groups in $[m] \setminus \sigma$. In particular, consider a paramterization $\nu : (\mathcal{S}_\rho, \mathcal{M}, [k]) \to \mathcal{R}^{1:m}$ defined as:

$$\nu(\mathbf{s}, \sigma, i) := \mathbf{r}^{1:m} \quad \text{such that} \quad \mathbf{r}^g = \begin{cases} \mathbf{e}_i & \text{if } g \in \sigma \\ \mathbf{s} & \text{o.w.} \end{cases} \tag{26}$$

i.e., $\nu$ assigns trivial confusion rates $\mathbf{e}_i$ on the groups in $\sigma$ and assigns $\mathbf{s} \in \mathcal{S}_\rho$ on the rest of the groups. Similar to the previous section, we first fix trivial classifier predicting class 1 for groups in $\sigma$ and constrain the rates for groups in $[m] \setminus \sigma$ to be on the sphere $\mathcal{S}_\rho$. Such a setup is governed by the parametrization $\nu(\cdot, \sigma, 1)$ in equation (26). Specifically, fixing $h^g(\mathbf{x}) = 1 \ \forall \ g \in \sigma$ would entail the metric in Definition 1 to be:

$$\Psi(\nu(\mathbf{s}, \sigma, 1); \mathbf{a}, \mathbf{B}, \lambda) = (1 - \lambda)\langle \mathbf{a} \odot (\mathbf{1} - \boldsymbol{\tau}^\sigma), \mathbf{s} \rangle + \lambda \langle \boldsymbol{\eta}^\sigma, |\mathbf{e}_1 - \mathbf{s}| \rangle + c_1, \tag{27}$$

where $\boldsymbol{\tau}^\sigma = \sum_{g \in \sigma} \boldsymbol{\tau}^g$ and $\boldsymbol{\eta}^\sigma = \sum_{u,v \in [m], v > u} \mathbb{1}\left[|\{u,v\} \cap \sigma| = 1\right] \mathbf{b}^{uv}$. Similar to the previous section, since $\mathbf{e}_i$'s are binary vectors, the sign of the absolute function wrt. $\mathbf{s}$ can be recovered. In particular, the metric amounts to:

$$\Psi(\nu(\mathbf{s}, \sigma, 1); \mathbf{a}, \mathbf{B}, \lambda) = \langle (1 - \lambda)\mathbf{a} \odot (\mathbf{1} - \boldsymbol{\tau}^2) + \lambda \mathbf{w}_1 \odot \boldsymbol{\eta}^\sigma, \mathbf{s} \rangle + c_1, \tag{28}$$

where $\mathbf{w}_1 := \mathbf{1} - 2\mathbf{e}_1$ and $c_1$ is a constant not affecting the responses. Notice that (27) and (28) are analogous to (19) and (20), respectively, except that $\boldsymbol{\tau}^2$ is replaced by $\boldsymbol{\tau}^\sigma$ and $\mathbf{b}^{12}$ is replaced by $\boldsymbol{\eta}^\sigma$. This is a linear metric in $\mathbf{s}$. We again the use the LPME procedure in line 8 of Algorithm 1, which outputs a normalized slope $\check{\mathbf{f}}^\sigma$ such that $\|\check{\mathbf{f}}^\sigma\|_2 = 1$, and thus we get an analogous solution to (21) as:

$$\lambda \boldsymbol{\eta}^\sigma = \mathbf{w}_1 \odot \left[ \left( \frac{(1 - \lambda)(1 - \tau^\sigma_{k-1})\bar{a}_{k-1} + \lambda \eta^\sigma_{k-1}}{\check{f}^\sigma_{k-1}} \right) \check{\mathbf{f}}^\sigma - (1 - \lambda)((\mathbf{1} - \boldsymbol{\tau}^\sigma) \odot \mathbf{a}) \right]. \tag{29}$$

In order to elicit entire $\boldsymbol{\eta}^\sigma$, we need one more linear relation such as (29). So, we now fix the trivial rates through trivial classifier predicting class $k$ for the groups in $\sigma$ i.e. fix $h^g(x) = k \ \forall \ \mathbf{x} \in \mathcal{X}$ if $g \in \sigma$, and thus $\mathbf{r}^g = \mathbf{e}_k$ for all groups $g \in \sigma$. For the rest of the groups, we constrain the confusion rates to again lie in the sphere $\mathcal{S}_\rho$ i.e. $\mathbf{r}^g = \mathbf{s}$ for $\mathbf{s} \in \mathcal{S}_\rho$ for all groups $g \in [m] \setminus \sigma$. Such a setup is governed by the parametrization $\nu(\cdot, \sigma, k)$ (26). The metric $\Psi$ in Definition 1 amounts to:

$$\Psi(\nu(\mathbf{s}, \sigma, k); \mathbf{a}, \mathbf{B}, \lambda) = (1 - \lambda)\langle \mathbf{a} \odot (\mathbf{1} - \boldsymbol{\tau}^\sigma), \mathbf{s} \rangle + \lambda \langle \boldsymbol{\eta}^\sigma, |\mathbf{e}_k - \mathbf{s}| \rangle + c_k. \tag{30}$$

Thus by running LPME procedure again in line 9 of Algorithm 1 results in $\tilde{\mathbf{f}}^{12}$ with $\|\tilde{\mathbf{f}}^{12}\|_2 = 1$. Using Remark 1, we extract the following relation between the $(k-1)$-th and $((k-1)^2 + 1)$-th coordinates:

$$\frac{\tilde{f}^\sigma_{k-1}}{\tilde{f}^\sigma_{(k-1)^2+1}} = \frac{(1 - \lambda)(1 - \tau^\sigma_{k-1})\bar{a}_{k-1} - \lambda \eta^\sigma_{k-1}}{(1 - \lambda)(1 - \tau^\sigma_{(k-1)^2+1})\bar{a}_{(k-1)^2+1} + \lambda \eta^\sigma_{(k-1)^2+1}}. \tag{31}$$

Combining equations (29) and (31), we have:

$$\sum_{u,v} \mathbb{1}\left[|\{u,v\} \cap \sigma| = 1\right] \tilde{\mathbf{b}}^{uv} = \boldsymbol{\gamma}^\sigma, \quad \text{where} \tag{32}$$

$$\boldsymbol{\gamma}^\sigma = \mathbf{w}_1 \odot \left[\delta^\sigma \mathbf{f}^\sigma - \hat{\mathbf{a}} \odot (\mathbf{1} - \boldsymbol{\tau}^\sigma)\right], \quad \delta^\sigma = \frac{2(1 - \tau^\sigma_{k-1})\hat{a}_{k-1}}{f^\sigma_{k-1}} \left[ \frac{\frac{(1 - \tau^\sigma_{(k-1)^2+1})\hat{a}_{(k-1)^2+1}}{(1 - \tau^\sigma_{k-1})\hat{a}_{k-1}} - \frac{\tilde{f}^\sigma_{(k-1)^2+1}}{\tilde{f}^\sigma_{k-1}}}{\left( \frac{f^\sigma_{(k-1)^2+1}}{f^\sigma_{k-1}} - \frac{\tilde{f}^\sigma_{(k-1)^2+1}}{\tilde{f}^\sigma_{k-1}} \right)} \right],$$

and $\tilde{\mathbf{b}}^{uv} := \lambda \mathbf{b}^{uv}/(1 - \lambda)$ is a scaled version of the true (unknown) $\mathbf{b}$, which nonetheless can be computed from (32).

By two runs of LPME algorithm, we can get $\boldsymbol{\gamma}^\sigma$ and solve (32). However, the left hand side of (32) does not allow us to recover the $\tilde{\mathbf{b}}$'s separately and provides only one equation. Let us denote the Equation (32) by $\ell^\sigma$ corresponding to the set $\sigma$. In order to elicit all $\tilde{\mathbf{b}}$'s we need a system of $M := \binom{m}{2}$ independent equations. This is easily achievable by choosing $M$ $\sigma$'s so that we get $M$ set of unique equations like (32). Let $\mathcal{M}$ be those set of sets. In most cases, pairing two groups to have trivial rates (through trivial classifiers) and rest of the groups to have rates from the sphere $\mathcal{S}$ will work. For example, when $m = 3$, fixing $\mathcal{M} = \{\{1, 2\}, \{1, 3\}, \{2, 3\}\}$ suffices. Thus, running over all the choices of sets of groups $\sigma \in \mathcal{M}$ provides the system of equations $\mathcal{L} := \cup_{\sigma \in \mathcal{M}} \ell^\sigma$ (line 10 in Algorithm 1), which is formally described as follows:

$$
\begin{bmatrix}
\Xi & 0 & \dots & 0 \\
0 & \Xi & \dots & 0 \\
\dots & \dots & \dots & \dots \\
0 & 0 & \dots & \Xi
\end{bmatrix}
\begin{bmatrix}
\tilde{\mathbf{b}}_{(1)} \\
\tilde{\mathbf{b}}_{(2)} \\
\dots \\
\tilde{\mathbf{b}}_{(q)}
\end{bmatrix}
=
\begin{bmatrix}
\boldsymbol{\gamma}_{(1)} \\
\boldsymbol{\gamma}_{(2)} \\
\dots \\
\boldsymbol{\gamma}_{(q)}
\end{bmatrix},
\tag{33}
$$

where $\tilde{\mathbf{b}}_{(i)} = (\tilde{b}_i^1, \tilde{b}_i^2, \cdots, \tilde{b}_i^M)$ and $\boldsymbol{\gamma}_{(i)} = (\gamma_i^1, \gamma_i^2, \cdots, \gamma_i^M)$ are vectorized versions of the $i$-th entry across groups for $i \in [q]$, and $\Xi \in \{0, 1\}^{M \times M}$ is a binary full-rank matrix denoting membership of groups in the set $\sigma \in \mathcal{M}$. For instance, for the choice of $\mathcal{M} = \{\{1, 2\}, \{1, 3\}, \{2, 3\}\}$ when $m = 3$ gives:

$$
\Xi =
\begin{bmatrix}
0 & 1 & 1 \\
1 & 0 & 1 \\
1 & 1 & 0
\end{bmatrix}.
$$

From technical point of view, one may choose any $\mathcal{M}$ such that the resulting group membership matrix $\Xi$ is non-singular. Hence the solution of the system of equations $\mathcal{L}$ is:

$$
\begin{bmatrix}
\tilde{\mathbf{b}}_{(1)} \\
\tilde{\mathbf{b}}_{(2)} \\
\dots \\
\tilde{\mathbf{b}}_{(q)}
\end{bmatrix}
=
\begin{bmatrix}
\Xi & 0 & \dots & 0 \\
0 & \Xi & \dots & 0 \\
\dots & \dots & \dots & \dots \\
0 & 0 & \dots & \Xi
\end{bmatrix}^{(-1)}
\begin{bmatrix}
\boldsymbol{\gamma}_{(1)} \\
\boldsymbol{\gamma}_{(2)} \\
\dots \\
\boldsymbol{\gamma}_{(q)}
\end{bmatrix}.
\tag{34}
$$

When we normalize $\tilde{\mathbf{b}}$, we get the final fairness violation weight estimates as:

$$
\hat{\mathbf{b}}^{uv} = \frac{\tilde{\mathbf{b}}^{uv}}{\sum_{u,v=1,v>u}^m \|\tilde{\mathbf{b}}^{uv}\|_2} \quad \text{for} \quad u, v \in [m], v > u.
\tag{35}
$$

Notice that, due to the above normalization, the solution is again independent of the true trade-off $\lambda$.

### C.3 Eliciting Trade-off $\lambda$; Part 3 in Figure 3 and line 12 in Algorithm 1

For ease of notation, let us construct a parametrization $\nu' : \mathcal{S}_\varrho^+ \to \mathcal{R}^{1:m}$:

$$
\nu'(\mathbf{s}^+) := (\mathbf{s}^+, \mathbf{o}, \dots, \mathbf{o}),
\tag{36}
$$

Using the parametrization $\nu'$ from (36), the metric in Definition 1 reduces to a linear metric in $\mathbf{s}^+$ as discussed in (15), i.e:

$$
\Psi(\nu'(\mathbf{s}^+) ; \mathbf{a}, \mathbf{B}, \lambda) = \langle (1 - \lambda)\boldsymbol{\tau}^1 \odot \mathbf{a} + \lambda \sum_{v=2}^m \mathbf{b}^{1v}, \mathbf{s}^+ \rangle + c.
\tag{37}
$$

We first show the proof of Lemma 1 and then discuss the trade-off elicitation algorithm (Algorithm 4).

*Proof of Lemma 1.* For simplicity, let us abuse notation for this proof and denote $\boldsymbol{\tau}^1 \odot \mathbf{a}$ simply by $\mathbf{a}$, $\sum_{v=2}^m \mathbf{b}^{1v}$ simply by $\mathbf{b}$, and $\mathcal{S}_\varrho^+$ simply by $\mathcal{S}$.

$\mathcal{S}$ is a convex set. Let $\mathcal{Z} = \{\mathbf{z} = (z_1, z_2) \,|\, z_1 = <\mathbf{a}, \mathbf{s}>, z_2 = <\mathbf{b}, \mathbf{s}>, \mathbf{s} \in \mathcal{S}\}$.

*Claim: $\mathcal{Z}$ is convex.*

Let $z, z' \in \mathcal{Z}$.

$\alpha z_1 + (1 - \alpha)z_1' = \alpha <\mathbf{a}, \mathbf{s}> + (1 - \alpha) <\mathbf{a}, \mathbf{s}'> = <\mathbf{a}, \alpha\mathbf{s} + (1 - \alpha)\mathbf{s}'>$

---
**Algorithm 4** Eliciting the trade-off $\lambda$
---

1: **Input:** Query space $\mathcal{S}_\varrho^+$, binary-search tolerance $\epsilon > 0$, oracle $\Omega^{\text{trade-off}}$
2: **Initialize:** $\lambda^{(a)} = 0, \lambda^{(b)} = 1$.
3: **while** $\left| \lambda^{(b)} - \lambda^{(a)} \right| > \epsilon$ **do**
4:    Set $\lambda^{(c)} = \frac{3\lambda^{(a)} + \lambda^{(b)}}{4}, \lambda^{(d)} = \frac{\lambda^{(a)} + \lambda^{(b)}}{2}, \lambda^{(e)} = \frac{\lambda^{(a)} + 3\lambda^{(b)}}{4}$
5:    Set $\mathbf{s}^{(a)} = \underset{\mathbf{s}^+ \in \mathcal{S}_\varrho^+}{\operatorname{argmax}} \langle (1 - \lambda_a)\boldsymbol{\tau}^1 \odot \hat{\mathbf{a}} + \lambda_a \sum_{v=2}^{m} \hat{\mathbf{b}}^{1v}, \mathbf{s}^+ \rangle$ using Lemma 2
6:    Similarly, set $\mathbf{s}^{(c)}, \mathbf{s}^{(d)}, \mathbf{s}^{(e)}, \mathbf{s}^{(b)}$.
7:    Query $\Omega^{\text{trade-off}}(\mathbf{s}^{(c)}, \mathbf{s}^{(a)}), \Omega^{\text{trade-off}}(\mathbf{s}^{(d)}, \mathbf{s}^{(c)}), \Omega^{\text{trade-off}}(\mathbf{s}^{(e)}, \mathbf{s}^{(d)})$, and $\Omega^{\text{trade-off}}(\mathbf{s}^{(b)}, \mathbf{s}^{(e)})$.
8:    $[\lambda^{(a)}, \lambda^{(b)}] \leftarrow$ *ShrinkInterval* (responses) using a subroutine analogous to the routine shown in Figure 5.
9: **Output:** $\hat{\lambda} = \frac{\lambda^{(a)} + \lambda^{(b)}}{2}$.

---

$\alpha z_2 + (1 - \alpha)z_2' = \alpha < \mathbf{b}, \mathbf{s} > +(1 - \alpha) < \mathbf{b}, \mathbf{s}' > = < \mathbf{b}, \alpha\mathbf{s} + (1 - \alpha)\mathbf{s}' >$

Since $\alpha\mathbf{s} + (1 - \alpha)\mathbf{s}' \in \mathcal{S}$, $\alpha z + (1 - \alpha)z' \in \mathcal{Z}$. Hence $\mathcal{Z}$ is convex.

*Claim:* The boundary of the set $\mathcal{Z}$ is a strictly convex curve with no vertices for $\mathbf{a} \neq \mathbf{b}$.

Recall that, the required function is given by:

$$\vartheta(\lambda) = \max_{\mathbf{z} \in \mathcal{Z}}(1 - \lambda)z_1 + \lambda z_2 + c \tag{38}$$

(i) Since the set $\mathcal{Z}$ is convex, every boundary point is supported by a hyperplane.

(ii) Since $\mathbf{a} \neq \mathbf{b}$, notice that the slope is uniquely defined by $\lambda$. Since the sphere $\mathcal{S}$ is strictly convex, the above linear functional defined by $\lambda$ is maximized by a unique point in $\mathcal{Z}$ (similar to Lemma 2). Thus, the the hyperplane is tangent at a unique point on the boundary of $\mathcal{Z}$.

(iii) It only remains to show that there are no vertices on the boundary of $\mathcal{Z}$. Recall that a vertex exists if (and only if) some point is supported by more than one tangent hyperplane in two dimensional space. This means there are two values of $\lambda$ that achieve the same maximizer. This is contradictory since there are no two linear functionals that achieve the same maximizer on $\mathcal{S}$.

This implies that the boundary of $\mathcal{Z}$ is strictly convex curve with no vertices. Since we are interested in the maximization of $\vartheta$, let us call this boundary as the upper boundary and denote it by $\partial\mathcal{Z}_+$.

*Claim:* Let $\upsilon : [0, 1] \to \partial\mathcal{Z}_+$ be continuous, bijective, parametrizations of the upper boundary. Let $\vartheta : \mathcal{Z} \to \mathbb{R}$ be a quasiconcave function which is monotone increasing in both $z_1$ and $z_2$. Then the composition $\vartheta \circ \upsilon : [0, 1] \to \mathbb{R}$ is strictly quasiconcave (and therefore unimodal with no flat regions) on the interval $[0, 1]$.

Let $S$ be some superlevel set of the quasiconcave function $\vartheta$. Since $\upsilon$ is a continuous bijection and since the boundary $\partial\mathcal{Z}_+$ is a strictly convex curve with no vertices, wlog., for any $r < s < t$, $z_1(\upsilon(r)) < z_1(\upsilon(s)) < z_1(\upsilon(t))$, and $z_2(\upsilon(r)) > z_2(\upsilon(s)) > z_2(\upsilon(t))$. (otherwise, swap $r$ and $t$). Since the boundary $\partial\mathcal{Z}_+$ is a strictly convex curve, then $\upsilon(s)$ must be greater (component-wise) a point in the convex combination of $\upsilon(r)$ and $\upsilon(t)$. Let us denote that point by $u$. Since $\vartheta$ is monotone increasing, then $x \in S$ implies that $y \in S$, too, for all $y \geq x$ componentwise. Therefore, $\vartheta(\upsilon(s)) \leq \vartheta(u)$. Since $S$ is convex, $u \in S$ and thus $\upsilon(s) \in S$.

This implies that $\upsilon^{-1}(\partial\mathcal{Z}_+ \cap S)$ is an interval; hence it is convex, which in turn tells us that the superlevel sets of $\vartheta \circ \upsilon$ are convex. So, $\vartheta \circ \upsilon$ is quasiconcave, as desired. This implies unimodaltiy, because a function defined on real line which has more than one local maximum can not be quasiconcave. Moreover, since there are no vertices on the boundary $\partial\mathcal{Z}_+$, the $\vartheta \circ \upsilon : [0, 1] \to \mathbb{R}$ is strictly quasiconcave (and thus unimodal with no flat regions) on the interval $[0, 1]$. This completes the proof of Lemma 1. $\qquad\square$

*Description of Algorithm 4:*[2] Given the unimodality of $\vartheta(\lambda)$ from Lemma 1, we devise the binary-search procedure Algorithm 4 for eliciting the true trade-off $\lambda$. The algorithm takes in input the query space $\mathcal{S}_\varrho^+$, binary-search tolerance $\epsilon$, an equivalent oracle $\Omega^{\text{trade-off}}$, the elicited $\hat{\mathbf{a}}$ from Section 4.1,

and the elicited $\hat{\mathbf{B}}$ from Section 4.2. The algorithm finds the maximizer of the function $\hat{\vartheta}(\lambda)$ defined analogously to (16), where $\mathbf{a}, \mathbf{B}$ are replaced by $\hat{\mathbf{a}}, \hat{\mathbf{B}}$. The algorithm poses four queries to the oracle and shrink the interval $[\lambda^{(a)}, \lambda^{(b)}]$ into half based on the responses using a subroutine analogous to *ShrinkInterval* shown in Figure 5. The algorithm stops when the length of the search interval $[\lambda^{(a)}, \lambda^{(b)}]$ is less than the tolerance $\epsilon$.

# D  Proof of Section 5

*Proof of Theorem 1.* Let $\| \cdot \|_\infty$ denote the $\ell$-infinity norm. We break this proof into three parts.

1. *Elicitation guarantees for the misclassification cost $\hat{\phi}$ (i.e., $\hat{\mathbf{a}}$)*

   Since Algorithm 1 elicits a linear metric using the $q$-dimensional sphere $\mathcal{S}$, the guarantees on $\hat{\mathbf{a}}$ follows from Theorem 2 of [21]. Thus, under Assumption 2, the output $\hat{\mathbf{a}}$ from line 1 of Algorithm 1 satisfies $\|\mathbf{a}^* - \hat{\mathbf{a}}\|_2 \leq O(\sqrt{q}(\epsilon + \sqrt{\epsilon_\Omega/\rho}))$ after $O\left(q \log \frac{\pi}{2\epsilon}\right)$ queries.

2. *Elicitation guarantees for the fairness violation cost $\hat{\varphi}$ (i.e., $\hat{\mathbf{B}}$)*

   We start with the definition of true $\boldsymbol{\gamma}$ (i.e. when all the elicited entities are true) from (32) and let us drop the superscript $\sigma$ for simplicity. Furthermore, let $\epsilon + \sqrt{\epsilon_\Omega/\rho}$ be denoted by $\epsilon$.

$$\boldsymbol{\gamma} = \mathbf{w}_1 \odot \left[\delta \breve{\mathbf{f}} - \mathbf{a} \odot (\mathbf{1} - \boldsymbol{\tau})\right] \quad \text{where } \delta = \frac{2(1 - \tau_{k-1})\bar{a}_{k-1}}{\breve{f}_{k-1}} \left[\frac{\frac{(1-\tau_{(k-1)^2+1})\bar{a}_{(k-1)^2+1}}{(1-\tau_{k-1})\bar{a}_{k-1}} - \frac{\tilde{f}_{(k-1)^2+1}}{\tilde{f}_{k-1}}}{\left(\frac{\breve{f}_{(k-1)^2+1}}{\breve{f}_{k-1}} - \frac{\tilde{f}_{(k-1)^2+1}}{\tilde{f}_{k-1}}\right)}\right].$$

   Let us look at the derivative of the $i$-th coordinate of $\boldsymbol{\gamma}$.

$$\frac{\partial \gamma_i}{\partial a_j} = \begin{cases} 0 & \text{if } j \neq i, j \neq k-1, j \neq (k-1)^2 + 1 \\ -\tau_i & \text{if } j = i \\ c_{i,1} & \text{if } j = k-1 \\ c_{i,2} & \text{if } j = (k-1)^2 + 1, \end{cases}$$

   where $c_{i,1}$ and $c_{i,2}$ are some bounded constants due to Assumption 2. Similarly, $\partial \gamma_i / \partial f_j$ is bounded as well due to the regularity Assumption 2. This means that $\gamma_i$ is Lipschitz in 2-norm wrt. $\mathbf{a}$ and $\mathbf{f}$. Thus,

$$\|\boldsymbol{\gamma} - \hat{\boldsymbol{\gamma}}\|_\infty \leq c_3 \|\mathbf{a} - \hat{\mathbf{a}}\|_2 + c_4 \|\breve{\mathbf{f}} - \hat{\breve{\mathbf{f}}}\|_2,$$

   for some Lipschits constants $c_3$ and $c_4$. From the bounds of Part 1 of this proof, we have:

$$\|\boldsymbol{\gamma} - \hat{\boldsymbol{\gamma}}\|_\infty \leq O(\sqrt{q}\epsilon).$$

   Recall the construction of $\tilde{\mathbf{b}}_{(i)}$ from (33). We then have from the solution of system of equations (34) that:

$$\tilde{\mathbf{b}}_{(i)} = \Xi^{-1} \boldsymbol{\gamma}_{(i)} \quad \forall\, i \in [q],$$

   where $\tilde{\mathbf{b}}_{(i)} = (\tilde{b}_i^1, \tilde{b}_i^2, \cdots, \tilde{b}_i^M)$ and $\tilde{\boldsymbol{\gamma}}_{(i)} = (\gamma_i^1, \gamma_i^2, \cdots, \gamma_i^M)$ are vectorized versions of the $i$-th entry across groups for $i \in [q]$. $\Xi \in \{0, 1\}^{M \times M}$ is a full-rank symmetric matrix with bounded infinity norm $\|\Xi^{-1}\|_\infty \leq c$ (here, infinity norm of a matrix is defined as the maximum absolute row sum of the matrix). Thus we have:

$$\|\tilde{\mathbf{b}}_{(i)} - \hat{\tilde{\mathbf{b}}}_{(i)}\|_\infty = \|\Xi^{-1}\boldsymbol{\gamma}_{(i)} - \Xi^{-1}\hat{\boldsymbol{\gamma}}_{(i)}\|_\infty = \|\Xi^{-1}(\boldsymbol{\gamma}_{(i)} - \hat{\boldsymbol{\gamma}}_{(i)})\|_\infty \leq \|\Xi^{-1}\|_\infty \|\boldsymbol{\gamma}_{(i)} - \hat{\boldsymbol{\gamma}}_{(i)}\|_\infty,$$

   which gives

$$\|\tilde{\mathbf{b}}_{(i)} - \hat{\tilde{\mathbf{b}}}_{(i)}\|_\infty \leq O(\sqrt{q}\epsilon).$$

Now, our final estimate is the normalized form of $\hat{\tilde{\mathbf{b}}}$ from (35), so the final error in the stacked version $vec(\mathbf{B})$ and $vec(\hat{\mathbf{B}})$ is:

$$\|vec(\mathbf{B}) - vec(\hat{\mathbf{B}})\|_\infty \leq O(\sqrt{q}\epsilon). \tag{39}$$

Since there are $q \times M$ entities in $vec(\mathbf{B})$, we have:

$$\|vec(\mathbf{B}) - vec(\hat{\mathbf{B}})\|_2 \leq O(\sqrt{qM}\sqrt{q}\epsilon) = O(mq\epsilon). \tag{40}$$

Due to elicitation on sphere and the oracle noise $\epsilon_\Omega$ as defined in Definition 4, we can replace $\epsilon$ with $\epsilon + \sqrt{\epsilon_\Omega/\rho}$ back to get the final bound on fairness violation weights as in Theorem 1.

3. *Elicitation guarantees for the trade-off parameter (i.e., $\hat{\lambda}$)*

   The metric for our purpose is a linear metric in $\mathbf{s}^+ \in \mathcal{S}_\rho^+$ with the following slope:

   $$\Psi(\nu'''(\mathbf{s}^+); \mathbf{a}, \mathbf{B}, \lambda) = \langle (1-\lambda)\boldsymbol{\tau}^1 \odot \mathbf{a} + \lambda \sum_{v=2}^m \mathbf{b}^{1v}, \mathbf{s}^+ \rangle. \tag{41}$$

   Since we elicit $\lambda$ through queries over a surface of the sphere, we pose this problem as finding the right angle (slope) defined by the true $\lambda$. Note that $\lambda$ is what we want to elicit; however, due to oracle noise $\epsilon_\Omega$, we can only aim to achieve a target angle $\lambda_t$. Moreover, we do not have true $\mathbf{a}$ and $\mathbf{B}$ but have only estimates $\hat{\mathbf{a}}$ and $\hat{\mathbf{B}}$. Thus we query proxy solutions always and can only aim to achieve an estimated version $\lambda_e$ of the target angle. Lastly, Algorithm 4 is stopped within an $\epsilon$ threshold, thus the final solution $\hat{\lambda}$ is within $\epsilon$ distance from $\lambda_e$. In total, we want to find:

   $$|\lambda - \hat{\lambda}| \leq \underbrace{|\lambda - \lambda_t|}_{\text{oracle error}} + \underbrace{|\lambda_t - \lambda_e|}_{\text{estimation error}} + \underbrace{|\lambda_e - \hat{\lambda}|}_{\text{optimization error}}.$$

   - optimization error: $|\lambda_e - \hat{\lambda}| \leq \epsilon$.
   - oracle error: Notice that the oracle correctly answers as long as $\varrho(1 - \cos(\lambda - \lambda_t)) > \epsilon_\Omega$. This is due to the fact that the metric is a 1-Lipschitz linear function, and the optimal value on the sphere of radius $\varrho$ is $\varrho$. However, as $1 - \cos(x) \geq x^2/3$, so oracle is correct as long as $|\lambda - \lambda_e| \geq \sqrt{3\epsilon_\Omega/\varrho}$. Given this condition, the binary search proceeds in the correct direction.
   - estimation error: We make this error because we only have access to the estimated $\hat{\mathbf{a}}$ and $\hat{\mathbf{B}}$ not the true $\mathbf{a}$ and $\mathbf{B}$. However, since the metric in (41) is Lipschitz in $\mathbf{a}$ and $\sum_{v=2}^m \mathbf{b}^{1v}$, this error can be treated as oracle feedback noise where the oracle responses with the estimated $\hat{\mathbf{a}}$ and $\hat{\mathbf{B}}$. Thus, if we replace $\epsilon_\Omega$ from the previous point to the error in $\hat{\mathbf{a}}$ and $\sum_{v=2}^m \hat{\mathbf{b}}^{1v}$, the binary search Algorithm 4 moves in the right direction as long as

   $$|\lambda_t - \lambda_e| \geq O\left(\sqrt{\frac{\|\mathbf{a} - \hat{\mathbf{a}}\|_2 + \sum_{v=2}^m \|\mathbf{b}^{1v} - \hat{\mathbf{b}}^{1v}\|_2}{\varrho}}\right) = O\left(\sqrt{mq(\epsilon + \sqrt{\epsilon_\Omega/\rho})/\varrho}\right),$$

   where we have used (40) to bound the error in $\{\hat{\mathbf{b}}^{1v}\}_{v=2}^m$.

Combining the three error bounds above gives us the desired result for trade-off parameter in Theorem 1.

$\square$

# E  Extended Experiments on Real-World Datasets; Ranking of Classifiers

One of the most important applications of performance metrics is evaluating classifiers, i.e., providing a quantitative score for their quality which then allows us to choose the best (or best set of) classifier(s). In this section, we discuss how the ranking of plausible classifiers is affected when a practitioner employs default metrics to rank (fair) classifiers instead of the oracle's metric or our elicited approximation.

Table 1: Dataset statistics; the real-valued regressor in *wine* and *crime* datasets is recast to 3 classes based on quantiles.

| Dataset | $k$ | $m$ | #samples | #features | group.feat |
|---------|-----|-----|----------|-----------|------------|
| default | 2 | 2 | 30000 | 33 | gender |
| adult | 2 | 3 | 43156 | 74 | race |
| wine | 3 | 2 | 6497 | 13 | color |
| crime | 3 | 3 | 1907 | 99 | race |

Table 2: Common (baseline) metrics usually deployed to rank classifiers.

| Name $\rightarrow$ | $\hat{\phi}\hat{\varphi}\hat{\lambda}$_a | $\hat{\phi}\hat{\varphi}\hat{\lambda}$_w | $\hat{\phi}\hat{\varphi}$_a | $\hat{\phi}\hat{\varphi}$_w | $\hat{\phi}$_a | $\hat{\phi}$_w | o_p | o_f |
|--------------------|------|------|------|------|------|------|------|------|
| $\hat{\mathbf{a}}$ | acc. | w-acc. | acc. | w-acc. | acc. | w-acc. | **a** | - |
| $\hat{\mathbf{B}}$ | acc. | w-acc. | acc. | w-acc. | elicit | elicit | - | **B** |
| $\hat{\lambda}$ | 0.5 | w-acc. | elicit | elicit | elicit | elicit | 0 | 1 |

We take four real-world classification datasets with $k, m \in \{2, 3\}$ (see Table 1). 60% of each dataset is used for training and the rest for testing. We create a pool of 100 classifiers for each dataset by tweaking hyperparameters under logistic regression models [28], multi-layer perceptron models [39], support vector machines [23], LightGBM models [26], and fairness constrained optimization based models [35]. We compute the group wise confusion rates on the test data for each model for each dataset. We will compare the ranking of these classifiers achieved by competing baseline metrics with respect to the ground truth ranking.

We generate 100 random oracle metrics $\Psi$. $\Psi$'s gives us the ground truth ranking of the above classifiers. We then use our proposed procedure FPME (Algorithm 1) to recover the oracle's metric. For comparison in ranking of real-world classifiers, we choose a few metrics that are routinely employed by practitioners as baselines (see Table 2). The prefixes (i.e. $\hat{\phi}$, $\hat{\varphi}$, or $\hat{\lambda}$) in name of the baseline metrics denote the components that are set to default metrics, and the suffixes (i.e. 'a' or 'wa') denote whether the assignment is done with *accuracy* (i.e. equal weights) or with *weighted accuracy* (weights are assigned randomly however maintaining the true order of weights as in $\Psi$). For example, $\hat{\phi}\hat{\varphi}\hat{\lambda}$_a corresponds to the metric where $\hat{\phi}, \hat{\varphi}, \hat{\lambda}$ are set to standard classification accuracy. Similarly, $\hat{\phi}$_w denote a metric where the misclassification cost $\hat{\phi}$ is set to weighted accuracy but both $\hat{\varphi}$ and $\hat{\lambda}$ are elicited using Part 2 and Part 3 of the FPME procedure (Algorithm 1), respectively. Assigning weighted accuracy versions is a commonplace since sometimes the order of the costs associated with the types of mistakes in misclassification cost $\phi$ or fairness violation $\varphi$ or preference for fairness violation over misclassification $\lambda$ is known but not the actual cost. Another example is $\hat{\phi}\hat{\varphi}$_a which corresponds to the metric where $\hat{\phi}, \hat{\varphi}$ are set to accuracy and only the trade-off $\hat{\lambda}$ is elicited using Part 3 of the FPME procedure (Algorithm 1). This is similar to prior work by Zhang et al. [53] who assumed the classification error and fairness violation known, so only the trade-off has to be elicited – however they also assume direct ratio queries, which can be challenging in practice. Our approach applies much simnpler pairwise preference queries. Lastly, o_p and o_f represent *only predictive performance* with $\lambda = 0$ and *only fairness* with $\lambda = 1$, respectively.

Figure 6 shows average NDCG (with exponential gain) [47] and Kendall-tau coefficient [43] over 100 metrics $\Psi$ and their respective estimates by the competing baseline metrics. We see that FPME, wherein we elicit $\hat{\phi}, \hat{\varphi}$, and $\hat{\lambda}$ in sequence, achieves the highest possible NDCG and Kendall-tau coefficient. Even though we make some elicitation error in recovery (Section 8), we achieve almost perfect results while ranking the classifiers. To connect to practice, this implies that when given a set of classifiers, ranking based on elicited metrics will align most closely to ranking based on the true metric, as compared to ranking classifiers based on default metrics. This is a crucial advantage of metric elicitation for practical purposes. In this experiment, baseline metrics achieve inferior ranking of classifiers in comparison to the rankings achieved by metrics that are elicited using the proposed FPME procedure. Figure 6 also suggests that it is beneficial to elicit all three components $(\mathbf{a}, \mathbf{B}, \lambda)$ of the metric in Definition 1, rather than pre-define a component and elicit the rest. For the *crime* dataset, some methods also achieve high NDCG values, so ranking at the top is good; however Kendall-tau coefficient is weak which suggests that overall ranking is poor. With the exception of the *default* dataset, the weighted versions are better than equally weighted versions in ranking. This is expected

Figure 6: Ranking performance of real-world classifiers by competing metrics.

because in weighted versions, at least order of the preference for the type of costs matches with the oracle's preferences.

## Footnotes

[1]The superscripts in Algorithm 2 denote iterates. Please do not confuse it with the sensitive group index.

[2]The superscripts in Algorithm 2 denote iterates. Please do not confuse it with the sensitive group index.