[Reviews · NeurIPS 2020]

Review 1

Summary and Contributions: Update: Thanks for your response. What you've sketched out in terms of showing the classifiers to users to compare makes sense. I think there's still some interesting questions there (e.g. how much easier is it for people to compare 2 classifiers based on their rates than to just write down their fairness metric), but I could see how that would be outside the scope of this work. I do still think that demonstrating the usefulness of this approach with at least synthetic users on real data is important though. You're probably right that the default metrics can arbitrarily mis-rank classifiers compared to some "true" metric if you design the true metric properly, but I wonder how often this is the case for plausible hypotheses of what the "true" human metric might be. It's hard to know how practically useful it is to involve users in this way without knowing this. --- This paper presents an approach to eliciting a performance metric for a classifier that combines both metrics based on misclassification and fairness. This metric could be elicited from a human by asking them to compare the statistics of different classifiers proposed by the algorithm in a linear number of queries. This can address the problem that fairness is a user-specific notion that can be difficult to quantify. This paper proposes a solution to this problem based on metric elicitation, and extends this approach to deal with the non-linearity required to solve this specific instantiation of the fairness metric problem. They describe the algorithm in detail, give guarantees about its performance, and demonstrate these guarantees in simulations.

Strengths: I really like the idea of eliciting people's ideas of what it means for a classifier to be fair in an application specific way. This paper presents a way of doing that that seems principled and reasonably likely to actually work in practice. I appreciated that they took the time to demonstrate how their fair performance metric can be mapped to existing fairness metrics with different settings of the parameters that are being elicited. This demonstrates that their method should be able to recover existing metrics when they are reasonable, as well as to perhaps discover new ones. I also appreciated that they incorporated the effect of oracle noise into their analysis since it's likely that there will be noise in the responses that people are able to give to this task.

Weaknesses: Since this paper addresses the problem of eliciting metrics from humans, I was disappointed to see that they didn't have any kind of even preliminary study with real users. This would be able to answer questions like whether people are able to give the kind of feedback requested by this algorithm, and how challenging it is for a user to directly specify their metric, or at least one close enough to get a good ranking of classifiers. Particularly with the first point, I would like to better understand what the task given to users would actually looks like. How are the classifiers that they must choose between presented to them? This seems like it could be challenging. I understand that user studies are hard and that they take time, but at least sketching out what the user aspect of this interaction would look like could shed some light on how reasonable it is to ask users to do this. I was also curious about how this would look with real data. Appendix E addresses this to some extent but I would really have liked to see it in the paper itself, and there were a couple of points from the analysis that I would have liked to see explored further. Figure 6 with the NDCG metric shows that most metrics do quite well for the crime dataset, and many do almost as well as FPME for other datasets. I see that the Kendall Tau metric is worse, but if NDCG looks at the top of the ranking as is mentioned in the paper, this seems like it would be the more relevant metric. The question I would most want to be able to answer with this kind of metric is whether it can accurately choose the best classifier from a set, so I'm not sure how important an accurate ranking in the middle of the distribution is. I'm also not sure how to interpret NDCG. Do the worse metrics actually choose "best models" according to the ranking they induce that are much worse than the "best model" chosen by FPME? I think it's important to understand to understand how useful this approach might be in practice. I also wanted to know how the random oracle metrics in the experiments where generated.

Correctness: As far as I can tell.

Clarity: I thought the paper was generally well written.

Relation to Prior Work: Yes

Reproducibility: No

Additional Feedback:


Review 2

Summary and Contributions: The manuscript proposes framework & algorithms for eliciting classification performance metric i.e. inferring an underlying, often implicit, performance metric of interest that combines fairness and classification accuracy from pairwise oracle queries. In this work fairness is defined wrt discrepancy in performances between different known (fixed) groups in the data. The framework extends existing work on metric elicitation in non-trivial and useful manner, and provides rigorous guarantees.

Strengths: Though the work builds on existing work on metric elicitation, the particular class of fairness performance metrics considered is significant and challenging (because of the non-linearity in the fairness violation part); the paper develops simple, intuitive and rigorously-justified ideas for the same.

Weaknesses: The paper satisfactorily develops and presents algorithms and guarantees for what it sets out to achieve. Some of the natural questions one would wonder, given the somewhat restrictive family of metrics, and the disconnect between oracle/expert's preferences and the assumptions made, are in fact discussed by the authors in conclusions as well as in broader impact. So I wouldn't hold anything in particular against this work. ----Update---- Thank you for the responses. ----------------

Correctness: The guarantees and the analysis look right to me, though I didn't thoroughly read through the proofs.

Clarity: Yes, I liked the writing in general and it was fairly easy to follow despite the paper being rather heavy on terms and notation. I thought some of the notation could be done away with (for example, I didn't see a reason for the \bar{} in the metrics in Section 4, why not just stick to the notation in Section 2?)

Relation to Prior Work: I'm not fully aware of the fairness literature, but the discussion on performance metric based references appears fairly thorough.

Reproducibility: Yes

Additional Feedback: The paper considers an important problem, given the increased attention and care to fairness, tied to increased deployment of machine learning models in societal and real-world decision-making. The manuscript builds on existing metric elicitation framework and techniques in non-trivial ways, makes fairly interesting and significant technical contributions. I liked the writing of the paper, flow and organization of ideas and the rigor in the work. I am in favor of accepting this work.


Review 3

Summary and Contributions: The paper discusses the problem of eliciting a metric including a fairness (expressed as the need to have balanced misclassifcation accross groups) component with it. The highly tehcnical results provide an efficient method that can accomodate finite sample and noisy feedbacks. The paper content looks legit, but is highly technical and require a lot of background knowledge/familiarity with the particular notations, making it poorly accessible outside some happy fews. As also indicate the large supplemental materials, this is typically a paper that would probably deserve a journal treatment rather than a packed A* conference one. And this is despite the fact that I am familiar with elicitation problems using pairwise preferences, and aware of fairness issues. Unfortunately for me and the authors (that will probably not get a lot from my review), the pandemic situation did not allow me to check the paper before the 13th (otherwise I would probably have sent it back).

Strengths: * A very timely topic * A quite general method for eliciting fairness performance metrics, that theoretically performs well under finite sample and noisy feedbacks (probably the most critical problem in elicitation procedures). * Some first experiments validating the feasability (if not applicability) of the proposed method on simulated decision makers.

Weaknesses: * A very high technicality, making it not very accessible to practicioners and persons that are not expert (maybe familiarity would be sufficient, but I would not bet on it) of the referenced works.

Correctness: It appears so, as far as I could get it, which is not very. I did not manage to run the provided code after trying for twenty minutes to download/reformat the default data set from the UCI repository, so I could not reproduce the empirical results (it would have been useful to provide a formatted data set with the code... it being 3MB, it should have been doable?)

Clarity: Well-written and organized, yes, without a doubt. Made accessible by trying to simplify the message as much as possible, I am not convinced of that. It would have been nice to have accessible (numerical) examples, if only in the supplemental materials. The notations are really many and there is a lot (really a lot) of background knowledge assumed. Note that while this is of course partly due to my own expertise, I also think that additional efforts could be made in this direction. I have read other (A* ML) papers that treat what I would consider equally complex issues, that were equally far from my expertise as this one, and that did a better job at accessibility.

Relation to Prior Work: * Yes

Reproducibility: No

Additional Feedback: I have very llittle, and I apologize for that, beyond trying to make their work accessible to a wider audience. I perceive this may be disappointing for the authors, at least as much as not being able to follow the paper was disappointing to me. I hope they get feedbacks from more expert readers than me.


Review 4

Summary and Contributions: This paper presents a method to learn fairness/performance preference from expert feed back in the multiclass group fairness setting with multiple sensitive groups. The authors present a number of theoretical results. Most notable is the algorithm to learn the fairness metric from an oracle, which they describe and provide a proof of correctness for. They also provide guarantees for robustness to noise in the expert feedback. Last, they provide brief empirical evidence of the correctness of their method by verifying that their algorithm recovers the fairness performance trade off in an artificial situation when the method is known.

Strengths: This paper presents a number of interesting and significant theoretical results. Eliciting fairness metrics from oracle preference is a compelling problem for fairness in machine learning and is in this way relevant to the NeurIPS community. In particular, this paper does a good job providing clear and compelling guidance through the proposed algorithm in the form of line by line guidance and justification. This aspect is welcome and makes the paper very readable. Overall, this paper comes off as a well written and high quality piece of work.

Weaknesses: Overall, this paper is clear and well written. However, the initial problem setup could be clarified in the introduction. In particular, figure 1 helps clarify the intended usage of the algorithm, but the figure is only cited in the text. Clarifying the intended usage of the algorithm in the text (if only briefly) would make the problem setup much clearer to readers new to the metric elicitation literature.

Correctness: I believe so.

Clarity: Yes, this paper is well written.

Relation to Prior Work: The paper is well situated in prior work. Though very different in nature, I think work has a strong connection with efforts that attempt to quantify user preferences of fairness notions in practice (i.e. https://arxiv.org/abs/1902.04783). Though not absolutely necessary, these may be worth discussing.

Reproducibility: Yes

Additional Feedback: Minor: I think there's a "The" missing on line 336. --- Thanks for the responses. My review remains unchanged after reading the author response.


Review 5

Summary and Contributions: This paper considers the question of how to choose a "fairness metric" for evaluating machine learning models. The authors consider the question through the lens of "metric elicitation", i.e., choosing a performance metric which "best" matches user preferences over accuracy and differences in prediction accuracy between groups. Experts are presented pairs of either classifiers or predictive rates and asked to indicate the one they prefer. The authors identify three challenges addressed by their work that are not addressed in previous works: complexity of jointly eliciting "fairness" and accuracy preferences; non-linear functions of predictive rates; the impact of the number of groups on query complexity.

Strengths: - Interesting extension of metric elicitation to take group-specific statistics and discrepancies into account. - Allows for different metric choices between different pairs of subgroups.

Weaknesses: - Equivalence of classifiers and rates: this is somewhat questionable from the perspective of fairness, as the classifier encodes significantly more information about subgroups. For example, suppose the classifier is determining whom to hire for a job. The "rate" of positive classification for qualified candidates from group A can be the same as group B, but individuals from group B are only classified positively if they live far away from the job location, and individuals from group A are only classified positively if they live close to the job location. This exactly allows the classifier to discriminate, and would be caught by an expert looking at the classifier, but not someone who only has access to the rate information for groups A and B (without location breakdown). It would be helpful to have some description of how this equivalence is acceptable from a fairness perspective, e.g., if there is an assumption we can make about the information in the rates for a specific setting. - Unclear if the method uncovers true fairness preferences: one question with this method is how well this can capture the user's understanding of fairness. It seems that one of the main inputs an expert user can provide is which groups are important to consider, but this method seems to require that the groups are specified up front, and that the expert cannot ask to change them. It's also not clear how intersections of groups are handled, e.g. gender and race. - Realistic model? It's not clear how realistic the model is, in particular the linear preferences, and how well the model adapts in cases in which the underlying metric or preference does not conform to the assumptions. (The authors may have a sense that it is realistic, in which case it would be helpful to give an example setting to help the reader better understand the assumptions.) It's also unclear if the set of classifiers allowed can only be those which achieve the same rates for each group. In a medical context, it may not be acceptable from a fairness perspective to select a classifier which behaves sub-optimally on one group in order to equalize rates across groups, and it was unclear from the stated results whether classifiers which achieve rates on group A that cannot be achieved on group B would be allowed. A more clear, complete statement of the assumptions of the model would be helpful in assessing in which settings this elicitation procedure may be relevant. - Handling multiple users' preferences? It would be interesting to see how the method can be adapted in the case of multiple users with potentially conflicting preferences. - It's hard to reconcile the lack of flexibility in the framework in taking expert feedback (e.g., specifying different subgroups, non-linear preferences, inconsistent preferences) with the statements in the "Broader Impact" setting.

Correctness: - Proofs were not included in the body of the paper. - Code correctness/reproducible results were not evaluated. - The experiment methodology seemed designed to evaluate how well the method uncovers a metric with very specific properties, but the paper claims that this method is practical, which would imply that it works well with human feedback.

Clarity: - The paper is very terminology heavy. In general, the terminology is explained, but a terminology table would be very helpful if space permits. - The text in the figures is quite small, particularly Figure 3. - Line 282 is unclear (maybe a missing phrase?)

Relation to Prior Work: This paper does a good job of explaining prior metric elicitation work, but it is a bit light on discussion of fairness definitions and in particular the weaknesses identified with definitions like equalized odds without consideration of many, intersecting subgroups. (See, e.g., Calibration for the (Computationally-Identifiable) Masses Úrsula Hébert-Johnson, Michael P. Kim, Omer Reingold, Guy N. Rothblum)

Reproducibility: Yes

Additional Feedback: This paper presents and interesting approach, and would be much improved by including a more complete example of the types of preferences the authors envision eliciting, and how this model is more practical than individual fairness or other metric or sub-group based approaches in that setting. It's also critical to discuss the assumptions of the model clearly and with an eye to practical issues including "interpretability" of the model, the influence of the presentation of the model and subgroups, etc on the users.


Review 6

Summary and Contributions: The paper considers the problem of eliciting the fairness metric in the multi-group and multi-class setting. The paper provides a theoretical framework to study the query complexity, where each query evaluates which classifier is preferred in terms of misclassification and fairness violations. Building upon previously studied metric elicitation technique that relies on the linearity of the performance, the paper reduces their non-linear performance metric into solving multiple instances of linear versions. They also perform a simulation of their algorithm, validating their theorems with randomly generated oracle metrics.

Strengths: The overall structure of the paper is sound, as they show how to incrementally solve different parts of the problem -- recovering the misclassification weights, the fairness violation weights, and then finally the trade-off parameter. And the authors do try to provide intuitions as to why each part works. They also empirically evaluate their approach, showing that they actually do recover the underlying metric.

Weaknesses: Although the paper indeed deals with a nonlinear function, it feels like it relies a lot upon the 'local linearity'; the nonlinear part, which is the fairness violation part, is just a sum of absolute differences, but each absolute difference is linear in essence. This 'local linearity' allows them to cleverly reduce to multiple calls to the linear queries. The only fairness notion that is considered in the paper is equalized odds, but I wonder whether it may be possible to handle other group fairness notions, such as positive/negative predictive value -- I think these values can be defined as weighted sum of r's, so there may be a way of reducing the problem to SLPME.

Correctness: Yes. I haven't checked all the details rigorously, but every major part all makes sense.

Clarity: Overall structure and the prose of the paper was pretty smooth. However, in my personal opinion, I felt that there were way too many superfluous notations such that it was hard to keep track of what each symbol represented; I didn't understand how certain notations help with the ease of reading the paper. I wish that the authors will remove some notations and actually just spell out the things (i.e. what the probability is instead of using kappas, zetas, etc). I believe that this will help with the readers not getting lost in the midst of all the minor details and not lose sight of the high level goal.

Relation to Prior Work: As mentioned above, they devise a clever way to use SLPME, which handles only linear metrics, in order to elicit their non-linear metric. And in their related work, they talk about how their paper relates to previous work.

Reproducibility: Yes

Additional Feedback: Some discussion about the advantage of recovering the metric, as opposed to solving the optimization problem without actually recovering the metric (i.e. Eliciting and Enforcing Subjective Individual Fairness https://arxiv.org/abs/1905.10660), may be helpful ------------------------------------------------------------------------------------------------------------------------------------------------ After the rebuttal: Overall, although as I pointed out locally, things are still linear, I like how this paper was able to cleverly handle the non-linearity. However, I think as the other reviewers have pointed out too, the paper will benefit much more if it actually tried to cut out some notations and actually fully spell the terms. And I think that kind of presentation improvement can't be just done with the notation table so I personally prefer that they try to spend more time with improving the overall clarity and accessibility of the paper. So, I'll keep my score at "6: marginally above the accpetance threshold".

[Author Response · NeurIPS 2020]

We thank the reviewers for their thorough reviews. As suggested, we will move some lesser used notations in the main paper to the appendix and include a terminology table. Below are responses to the main comments.

**Reviewer 1. User study:** The oracle can be an expert, a group of experts, or an entire user population. Our framework can be applied by posing classifier comparisons directly via interpretable learning techniques [36, 10] or via A/B testing [39] (section 8, point 3). For example, for internet-based applications, one may perform A/B testing by deploying two classifiers A and B with two different sub-populations of users and use their level of engagement to decide which of the two classifiers is more preferred. For other applications, we may present to the user, visualizations of the predictive rates for two different classifiers, along with textual explanations, and have the user provide pairwise feedback. See e.g. Figure 3 in ref. [45] in the paper, or Figure 2 in ref. [A] below for intuitive ways to visualize rates.

**Real-data experiments:** We'll certainly move the experiments in Appendix E to the main text, given the additional page. We note that most baselines use some form of elicitation or prior knowledge. We wish to make two points in clarification: (a) even for this simple example, the results show that some kind of elicitation is usually preferable to no elicitation, even when measured by NDCG, (b) perhaps more importantly, since we have generated the oracle's metric randomly according to Definition 1, we may have picked a case that is too benign. It is not difficult to construct realistic settings where a default metric orders the classifiers arbitrarily different at the top from the true oracle's metric.

**Reviewer 2:** We thank the reviewer for the positive feedback. We'll simplify the suggested notations. With regard to mismatch between oracle preferences and the metric, please see the response to Reviewer 6.

**Reviewer 3.** While we have dedicated sub-sections on the problem statement and background material in Section 2, we understand why the reviewer might have found it to be insufficient. To make our work more accessible to non-expert readers, we will use the additional page for (a) adding an end-to-end numerical example, (b) elaborating on the metric selection problem with the intended use-cases around Figure 1, and (c) giving more intuition about each elicitation step.

**Reviewer 5.** Thanks for the positive feedback. As suggested, we'll clarify the intended use-cases early on with more text around Figure 1 and include the additional reference as well. Thanks for the helpful suggestions.

**Reviewer 6. Equivalence of classifiers and rates:** Our goal is to elicit *group-fairness metrics* such as equal opportunity [14] that can be expressed as a function of group-specific rates of a classifier. Since the metrics we consider depend on a classifier only through its rates (lines 46, 116), comparing two classifiers on these metrics is equivalent to comparing their rates. The concern the reviewer raises is true in general with any group-based notion of fairness and is not specific to our setup. While there are alternate definitions of fairness that look at individual outcomes, and some early work on eliciting individual fairness metrics [18, 28], these are tangential to the current paper (see Section 7 for details).

**Assumption of fixed groups:** This is a standard assumption in the fairness literature (see e.g. [2, 4, 9, 14, 21, 24, 27]). We agree that eliciting the fairness metric when the groups change or are unknown is an important and practical question, but beyond the scope of this paper. Moreover, we can adapt the standard practice of treating each intersection as a separate subgroup to handle intersections of groups. Other ways of handling overlapping subgroups for eliciting group-fair metrics is a promising future direction. Thanks for the additional reference. We'll add a discussion on this.

**Mismatch between preferences and metric.** The class of metrics we handle is fairly broad and includes many common classification metrics such as weighted accuracy, popular group fairness metrics such as equal opportunity, and combinations of those. Indeed our approach is able to both recover existing metrics and discover new ones. Moreover, there is precedent in various application domains to modeling user preferences with a parametric form [e.g. B-D], including prior work on metric elicitation [16, 17]. We thus see our work as a significant first step towards eliciting more general classes of fairness metrics. On the question about feasibility, we allow all feasible rates, i.e. for which there exists a classifier; for elicitation, we exploit a subset of these rates (e.g. sphere $\mathcal{S}_\rho$) with specific feasibility properties.

**Multiple user preferences.** Our approach is robust to noise in the preference feedback (see Section 5). We can handle many common noise models including the one described in Definition 4, and a model where the feedback is only probably correct. One way to handle multiple conflicting preferences is to simply aggregate them into a single feedback (e.g. with a majority vote), and use the aggregated feedback for elicitation.

**Reviewer 7. Local-linearity:** The reviewer is correct that the metrics we consider are piece-wise linear functions of rates, but we needed to substantially extend prior work on linear metric elicitation to be able to exploit this structure. A key challenge was to select queries so that we can jointly elicit the performance and fairness violation by either zeroing out or linearizing these terms. Handling more general non-linear functions is a promising direction for future work.

**Metrics:** In addition to equalized odds, our approach can recover many common fairness metrics such as equal opportunity, error-rate balance, etc. (see lines 97–101), and can also be extended to handle demographic parity [2]. It would be interesting to explore metrics such as positive predictive value that are fractional-linear functions of rates [17].

**Optimization w/o elicitation:** Thanks for the interesting pointer. We'll add a note on this.

**Added References:** [A] "Visualization of Confusion Matrix for Non-Expert Users" [B] Abbasi-yadkori et al. "Improved Algorithms for Linear Stochastic Bandits" [C] Valkenhoef et al. "Entropy-optimal weight constraint elicitation with additive multi-attribute utility models" [D] Abeel et al. "Apprenticeship Learning via Inverse Reinforcement Learning"

[Meta-Review · NeurIPS 2020]

The following meta-review is based on reading all reviews, the author response, the discussion, and the submission (though not the supplementary material). All reviewers agree that this paper studies an important problem: how to elicit fair performance metrics from user (or oracle) feedback. The paper proposes a non-trivial extension of the recently introduced Metric Elicitation framework as an approach to this task. Overall this was assessed to be a borderline submission. While the paper is strong in terms of its technical formalism, and the presentation is very precise, it is also very challenging for all but the most dedicated reader to follow along. This is due in large part to two factors: (1) the abundance of notation introduced and relied upon in lieu of more descriptive terms throughout; and (2) the emphasis on technical precision over more reader-friendly narrative style prose. Furthermore, while the main contribution of the paper is to the fairness literature, the work lacks a meaningful discussion of how the work is situated within the broader work on algorithmic fairness. I second here all of the points raised by R6. I encourage the authors to focus their revision on (1) making the paper more accessible; and (2) providing a more in-depth, "practitioner oriented" discussion of the contributions and limitations of the work within the algorithmic fairness literature. The paper would be made much stronger, not weaker, if the authors clarified, for instance, how reducing classifiers to rates may sweep under the rug distinctions that users feel are important (again, see R6), and how the assumptions/inputs to the method constrain what dimensions of fairness can and cannot be learned about. To be clear: I see the current paper as one that other researchers may be interested in building upon. However, the current presentation of the paper is likely to severely limit its audience and ultimate impact.